

# The importance of tree demography and root water uptake for modelling the carbon and water cycles of Amazonia

Emilie Joetzjer[1,2], Fabienne Maignan[1], Jérôme Chave[2], Daniel Goll[1], Ben Poulter[3], Jonathan Barichivich[1,4], Isabelle Maréchaux[2,5], Sebastiaan Luyssaert[1,6], Matthieu Guimberteau[1,7], Kim Naudts[1,8], Damien Bonal[9], Philippe Ciais[1]

[1]Laboratoire des Sciences du Climat et de l'Environnement, LSCE-IPSL (CEA-CNRS-UVSQ), 91190 Gif-sur-Yvette, France
[2]Laboratoire Evolution et Diversité Biologique, UMR 5174, Université Paul Sabatier, CNRS, IRD, 31400 Toulouse, France
[3]NASA Goddard Space Flight Center, Biospheric Sciences Laboratory, Greenbelt, MD, USA
[4]Instituto de Conservación, Biodiversidad y Territorio, Universidad Austral de Chile, Valdivia, Chile, and Center for Climate and Resilience Research, Santiago, Chile
[5]AMAP, Université de Montpellier, IRD, CIRAD, CNRS, INRA, 34000 Montpellier, France
[6]Vrije Universiteit Amsterdam, Faculty of Science, 1081 HV, The Netherlands.
[7]UMR 7619 METIS, Sorbonne Universités, UPMC, CNRS, EPHE, 4 place Jussieu, 75005 Paris, France
[8]Max Planck Institute for Meteorology, Bundesstraβe. 53, 20146 Hamburg, Germany
[9]Université de Lorraine, AgroParisTech, INRA, UMR Silva, 54000 Nancy, France

*Correspondence to*: Emilie.joetzjer@lsce.ipsl.fr

**Abstract.** Amazonian forest plays a crucial role in regulating the carbon and water cycles in the global climate system. However, the representation of biogeochemical fluxes and forest structure in dynamic global vegetation models (DGVMs) remains challenging. This situation has considerable implications for modelling the state and dynamics of Amazonian forest. To address these limitations, we present an adaptation of the ORCHIDEE-CAN DGVM, a second-generation DGVM that explicitly models tree demography and canopy structure with an allometry-based carbon allocation scheme and accounts for hydraulic architecture in the soil-stem-leaf continuum. We use two versions of this DGVM: the first one (CAN) includes a new parameterization for Amazonian forest; the second one (CAN-RS) additionally includes a mechanistic root water uptake module, which models the hydraulic resistance of the water transfer from soil pores to roots. We compared the results with the simulation output of the "big-leaf" standard version of the ORCHIDEE DGVM (TRUNK) and with observations of turbulent energy and $CO_2$ fluxes at flux tower locations, of carbon stocks and stand density at inventory plots and observation-based models of photosynthesis (GPP) and evapotranspiration (LE) across the Amazon basin. CAN-RS reproduced observed carbon and water fluxes and carbon stocks as well as TRUNK across Amazonia, both at local and at regional scales. In CAN-RS, water uptake by tree roots in the deepest soil layers during the dry season significantly improved the modelling of GPP and LE seasonal cycles, especially over the Guianan and Brazilian Shields. These results imply that explicit coupling of the water and carbon cycles improves the representation of biogeochemical cycles in Amazonia and their spatial variability. Representing the variation in the ecological functioning of Amazonia should be the next step to improve the performance and predictive ability of new generation DGVMs.

## 1 Introduction

Amazonian rainforests store approximately half of the world's tropical forest carbon stock (Baccini et al., 2012) and play a crucial role in global water, energy and carbon cycling (Eltahir and Bras, 1994; Werth and Avissar, 2002). The resilience and





resistance of these forests to climate change is of great concern, especially since a significant portion of Amazonia will likely experience longer and drier dry seasons by the end of the 21st century (Boisier et al., 2015; Joetzjer et al., 2013). Future changes in the rate of carbon sequestered by Amazonia could potentially lead the global climate system to a critical tipping point (Ahlström et al., 2017; Lenton et al., 2008; Nobre and Borma, 2009), and trigger positive carbon cycle climate

feedbacks from forest dieback. Yet, large uncertainties impede the production of robust future projections of changes in net carbon uptake over Amazonia (Arora et al., 2013; Jones et al., 2013; Poulter et al., 2010) – current model projections range from no change, or an increase in tree biomass production (Cox et al., 2013; Huntingford et al., 2013; Rammig et al., 2010), to large-scale Amazonian dieback (Cox et al., 2004; Good et al., 2011).

An analysis of variance on simulation outputs from 12 Earth System models (ESM) showed that uncertainties in projections of terrestrial carbon uptake are primarily driven by model structure (Lovenduski and Bonan, 2017). These uncertainties arise from both the atmospheric (Ahlström et al., 2012) and the land surface components (Booth et al., 2012; Sitch et al., 2015). In land models (dynamic global vegetation models, or DGVMs) large sources of uncertainty include the vegetation response to droughts (Restrepo-Coupe et al., 2016), and tree demographic processes (Fisher et al., 2010; Rödig et al., 2018). Most

DGVMs simulate the effect of water shortage on plant functioning by lowering leaf gas exchange rates using a multiplicative water stress factor that depends on soil moisture (Christoffersen et al., 2014) and by including atmospheric water stress from increased vapour pressure deficit in their parameterization of stomatal conductance. With this simplification, models typically fail to capture tropical carbon and water flux seasonality (Poulter et al., 2009; Restrepo-Coupe et al., 2016), and vegetation response to drought (Joetzjer et al., 2014; Powell et al., 2013). A few global DGVMs have recently adopted a

more explicit representation of the soil-plant-atmosphere water column [e.g., *Bonan et al.*, 2014; *Christoffersen et al.*, 2014], but much research is still needed to fully model these processes.

In most DGVMs, water availability in the root zone is quantified using the root biomass-weighted or root profile-weighted sum of soil layer moisture. Yet, this model structure overlooks the observation that soil-to-root water flow depends on soil

and root hydraulic properties, which vary in time and space (Sperry et al., 2002). A prevailing assumption is that the upper soil layers, with higher root biomass, contribute more to soil water uptake. This however overlooks the fact that tree water potentials preferentially equilibrate with the wettest part of the soil (Schmidhalter, 1997), a process controlled not only by the density of root tissue but also by the soil-to-root resistance. In turn, the soil-to-root resistance is non-linearly related to soil water content (Gardner, 1960). Overall, this approach could lead to an overestimation of the water stress experienced by

trees.

First-generation "big-leaf" DGVMs are progressively being superseded by second generation DGVMs (2gDGVM). This new generation of models is partly inspired by individual plant-based and forest stand models (e.g., [*Fyllas et al.*, 2014; *Fischer et al.*, 2016; *Maréchaux and Chave*, 2017]), and they explicitly represent forest dynamics via tree demography




(cohort-based) and vertical competition for light. 2gDGVMs are currently the state-of-the-art tools to understand vegetation response to climatic perturbations over large spatial scales (Fisher et al., 2018), although they come with increasingly complex parameterizations.

This study explores the relative contributions of tree demographic, canopy structure and hydraulic processes on the Amazonian carbon and water cycles. We present several improvements to the ORCHIDEE DGVM (Organizing Carbon and Hydrology in Dynamic Ecosystems). The original version, henceforth called TRUNK, was described by *Krinner et al.,* [2005]. Here, we have used a recent release, updated for the CMIP6 exercise (Peylin et al., *in prep* ; https://orchidee.ipsl.fr/ ). The second model version, ORCHIDEE-CAN (for Canopy, henceforth abbreviated to CAN), was first described by *Naudts*

*et al.* [2015] but was not parameterized for Amazonia. CAN includes (i) an explicit tree demography with size-dependent carbon allocation, (ii) a vertical discretization of the radiative transfer and energy budget calculations (Ryder et al., 2016), and (iii) an explicit representation of tree hydraulic architecture, based on the scheme proposed by *Hickler et al.,* [2006]. This study evaluates CAN's performance with a parameterization for humid tropical forest. Finally, we implemented an improved representation of the root water uptake process based on the work of *Williams et al.,* [2001]. We called this new

version ORCHIDEE-CAN-RS (for Root-Soil, henceforth abbreviated to CAN-RS). It accounts for the layer-to-layer heterogeneity in soil-to-root resistance to simulate the pattern of plant water uptake.

By comparing simulations by these three versions of the same DGVM over Amazonia, our results shed light on critical processes whose explicit representation would help to improve the performance of 2gDGVMs, and enhance their predictive ability on the fate of the largest tropical forest on Earth.

## 2. Methods

### 2.1 Model description and experimental design

#### 2.1.1 General model description

ORCHIDEE is a process-based ecosystem model first described by *Krinner et al.,* [2005]. It represents energy, water, and carbon exchanges within the soil-plant-atmosphere continuum using a big-leaf approach. Carbon assimilation is based on the leaf-scale equation of *Farquhar et al.,* [1980] for C3 plants and is assumed to scale from leaf to canopy with APAR

(absorbed photosynthetically active radiation) decreasing exponentially with leaf area index (LAI), according to the big-leaf approximation. Stomatal conductance is proportional to the product of net $CO_2$ assimilation with atmospheric relative humidity divided by atmospheric $CO_2$ concentration in the canopy [*Ball et al.*, 1987]. Evapotranspiration (ET) is the sum of evaporation from bare soil, evaporation of water intercepted by the canopy, and transpiration. Transpiration is controlled by



the stomatal conductance, which is modelled as a function of water availability in the soil column and of a fixed root density profile (de Rosnay and Polcher, 1998).

The CAN version of ORCHIDEE (McGrath et al., 2016; Naudts et al., 2015) replaced the big-leaf approach by a dynamic three-dimensional representation of the forest canopy. Forest tree demography, including recruitment, is simulated by

5    distributing stand-level net primary productivity (NPP) to a user-defined number of diameter classes following the size-dependent allocation rule of *Deleuze et al.*, [2004], as originally implemented by *Bellassen et al.*, [2010]. Mortality due to competition is based on a relationship between biomass and diameter, i.e., self-thinning (Reineke, 1933). This process has been widely reported for temperate and boreal forests, but it has also been observed, albeit with a larger noise, in humid tropical forests (Kohyama, 1992; Phillips et al., 2002; Pillet et al., 2017). Additionally, because actual transpiration is limited

10   by the amount of water the plant can transport from the soil to its leaves, this is calculated as a function of the ratio of the water potential difference between soil and leaves. This procedure accounts for the total hydraulic resistance of the water pathway from roots to sapwood and leaves, described by *Hickler et al.*, [2006] and *Naudts et al.*, [2015]. CAN was originally parameterized and evaluated for mid-latitude forests (Naudts et al., 2015). The main adaptations made to CAN for tropical forests are presented below, with minor changes listed in the Supplementary Information.

### 2.1.2 ORCHIDEE-CAN: self-thinning and recruitment scheme

In CAN, competition for light among trees is simulated through self-thinning. The maximum number of trees ($N_{max}$) depends on the mean stand diameter ($D_g$ (m), Fig. 1a) as follows:

$$N_{max} = \left(\frac{D_g}{\alpha}\right)^{1/\beta} \tag{1}$$

with the parameters $\alpha = 1100$ (m) and $\beta = -0.57$ estimated for tropical forests using publicly available plot-level data from the RAINFOR forest inventory network (Brienen et al., 2015).

In unmanaged tropical forests, the mortality of old trees takes place in parallel with the recruitment of young trees. To account for this natural plant regeneration, we implemented a recruitment scheme where the number of recruited trees ($N_{recruits}$ per hectare and per year, Fig. 1b) is a function of the LAI using the following equation:

$$N_{recruits} = 100 \exp(- 0.15 \, LAI) \tag{2}$$





This parameterization assumes that the number of recruits depends on mean-stand LAI. Note that in this version recruitment rate does not depend on the rate of canopy gap formation, in spite of the demonstrated importance of canopy openings for the regeneration of natural forests.

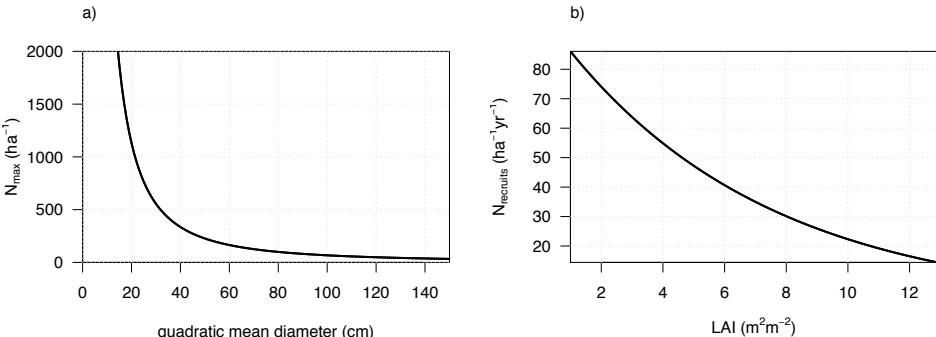

**Figure 1. (a) self-thinning equation and (b) recruitment scheme for tropical forests in CAN**

### 2.1.3 Implementing a dynamic root scheme

In CAN, the soil water potential in the rooting zone ($\Psi_{rz}$, MPa) is calculated as the weighted sum of the soil water potential

per layer ($\Psi_s$, MPa), weighted by the relative proportion of root biomass in each layer $d_{root}$. An additive tuning factor ($m_\psi$) accounts for missing processes, such as the hydraulic resistance at the soil-root interface (Naudts et al., 2015) :

$$\Psi_{rz} = \sum_{l=1}^{L}[\Psi_s(l)d_{root}(l)] + m_\psi \tag{3}$$

where $L$ is the number of layers (L=12).

$\Psi_s$ is calculated for each soil layer $l$ and depends on the layer volumetric water content ($\Theta(l)$ m$^3$ m$^{-3}$) following the Mualem - van Genuchten model (van Genuchten, 1980; Mualem, 1976):

$$\Psi_s(l) = \min\left(\frac{1}{k_{av}} \left(\left(\frac{SWC(l)-\Theta_r}{\Theta_r-\Theta_s}\right)^{-\frac{1}{k_{mv}}} - 1\right)^{\frac{1}{k_{nv}}}; -5\right) \tag{4}$$

where $SWC(l)$ is soil water content in layer $l$; $\Theta_r$ and $\Theta_s$ (m$^3$ m$^{-3}$) are the residual and saturated SWC, respectively; and $k_{av}, k_{mv}$ and $k_{nv}$ are the van Genuchten parameters. These parameters are texture-dependent (see Table S1). $\Psi_s$ cannot be





lower than the soil water potential for hygroscopic water (-5 MPa) (Larcher, 2003).

The use of root biomass-weighted sum of soil layer moisture in Eq. (3) ignores the dependence of soil-to-root water flow on soil and root hydraulic properties. Besides, the use of $m_\psi$ often leads to incorrect positive hydraulic potentials. We therefore

implemented a different computation of $\Psi_{rz}$, whereby $\Psi_s$ is weighted by $E_{max}(l)$, the maximum amount of water (mmol m$^2$ s$^{-1}$) that can be absorbed by the roots in each layer, which itself depends on the soil-to-root resistance $R_{sr}$ (MPa s mmol$^{-1}$ m$^{-2}$) and on a minimum root water potential $\Psi_{root,m}$ (MPa) (Duursma and Medlyn, 2012; Fisher et al., 2006; Williams et al., 2001). Replacing Eq. (3) by the following equation in CAN leads to the version hereafter called CAN-RS.

$$\Psi_{rz} = \frac{\sum_1^L \Psi_s(l)\, E_{max}(l)}{\sum_1^L E_{max}(l)} \text{ with } E_{max}(l) = [\Psi_s(l) - \Psi_{root,m}]/R_{sr}(l) \qquad (5)$$

$\Psi_{root,m}$ is a parameter set at -3 MPa (Duursma and Medlyn, 2012). The soil-to-root resistance $R_{sr}$ estimates the effective pathlength for water transport from the soil matrix to the root surface (Gardner, 1960), and is computed as follows:

$$R_{sr}(l) = \frac{\ln\left(\frac{r_s(l)}{r_r}\right)}{2\,\pi\, l_r(l)\, G_{soil}(l) \Delta_D(l)} \qquad (6)$$

Here, $l_r$ (m$^{-2}$) is the root length per unit of soil volume, and is a function of the specific root length (SRL), with SRL set at 10 m g$^{-1}$ (Metcalfe et al., 2008), and of the fine root biomass density per layer ($Biomass_{froots}(l)$, in g m$^{-3}$): $l_r(l) = Biomass_{froots}(l)\, SRL$; $r_s$ (m) is one-half of the mean distance between roots, computed following (Newman, 1969):

$$r_s = \left(\frac{1}{\pi\, l_r(l)}\right)^{0.5} \qquad (7)$$

and $r_r$ (m) is the mean fine root radius, set at 0.29 10$^{-3}$ m (Bonan et al., 2014); $G_{soil}$ (mmol m$^{-1}$ s$^{-1}$ MPa$^{-1}$) is the saturated hydraulic conductivity for the soil (see section 2.1.3). In CAN, $Biomass_{froots}$ is calculated following the allocation scheme relying on the pipe model theory (Shinozaki, 1964; Sitch et al., 2003) and it is vertically discretized per soil layer by

multiplying by $d_{root}(l)$.

### 2.1.3 Soil characteristics

In all three versions of ORCHIDEE considered in this study, the relationships between saturated hydraulic conductivity,

volumetric water content, and matrix potential are described by the Mualem–van Genuchten model (van Genuchten, 1980;



Mualem, 1976), using the parameters estimated by *Carsel and Parrish*, [1988] for the 12 soil texture classes of the United States Department of Agriculture (USDA) classification.

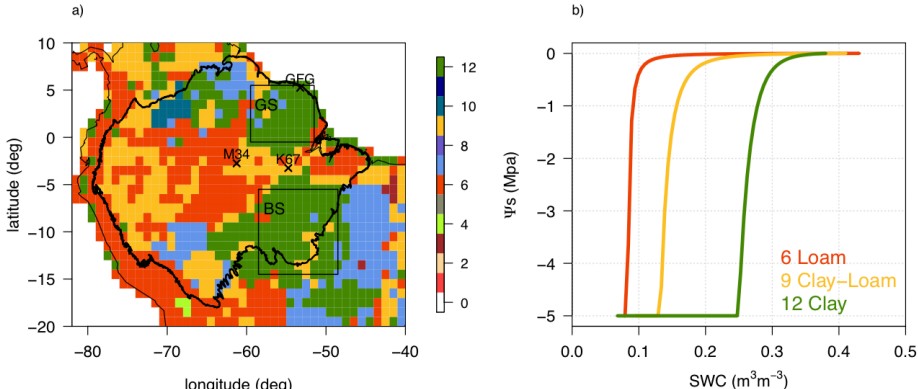

**Figure 2. (a) USDA soil types interpolated at 1-degree resolution over the Amazon used as a forcing to ORCHIDEE, where GS and BS squares represent the Guianan and Brazilian Shields respectively; and (b) the soil water retention curves ($\Psi_s$ versus SWC) predicted by the Mualem-van Genuchten equation (Eq. 4) for the three dominant USDA soil classes in Amazonia. Parameter values are given in the SI (Table S1).**

The spatial heterogeneity of soil structure in Amazonia is related to the geology of the area with old, highly weathered soils (Precambrian substrates) over the Brazilian and Guianan Shields contrasting with the much younger Cenozoic geology of the Andes and western Amazonia (Quesada et al., 2011). This is reflected in the USDA map of soil types, with mainly clayey (12) soil type over the shields, and loam (6) and clay-loam (9) over the rest of Amazonia (Fig. 2a). The Mualem-van Genuchten equation (Eq. 4) implies lower water availability for plants in clayey soils than for those in loam or clay-loam soils, at a given soil water content (i.e., more negative values $\Psi_s$ for the same SWC; Fig. 2b).

### 2.1.4 Simulations

To investigate the effect of both hydraulic processes and model structure on the simulated forest dynamics of Amazonia, we compared outputs from the TRUNK version as used in the Sixth Model Intercomparison Project (CMIP6), the CAN (v2290) version parameterized for tropical forests, and CAN-RS. All three versions were run using 13 plant functional types (PFTs) and the multi-layer diffusion scheme [*de Rosnay et al.*, 2002] considering a 4-metre soil depth and 12 soil layers (Campoy et al., 2013).

Firstly, simulations were performed at three sites across Brazil and French Guiana for which eddy-covariance measurements were available (da Rocha, 2004): Santarem KM67 (K67), Manaus KM34 (M34) (da Rocha et al., 2009) and Paracou (GFG) (Bonal et al., 2008). The evergreen tropical forest PFT cover was used. All simulations used hourly local meteorological





forcing. Each site corresponds to one of the major soil texture classes according to USDA soil classification (Fig. 2 and Table 1). Secondly, regional historical simulations were performed at 1-degree spatial resolution over the Amazonian forest using up-scaled gridded climate forcing data from CRUNCEP, which combine monthly data from the Climate Research Unit (CRU) and 6-hourly fields from the National Center for Environmental Prediction (NCEP) (Wei et al., 2014) (Table 1). All

simulations started from a semi-analytical spin-up (Lardy et al., 2011) to equilibrate carbon and hydrological variables by recycling climate data from 1981 to 2000 under a constant $CO_2$ concentration set to 370 ppm.

**Table 1. Summary of the simulations. All four simulations were run with the TRUNK, CAN and CAN-RS versions.**

|  | Simulation name | Soil type | USDA | Meteorological data | Period |
|---|---|---|---|---|---|
| **local** | K67 | clay-loam | 9 | In situ meteorological measurements (hourly) | 2002-2004 |
|  | M34 | loam | 6 |  | 2003-2005 |
|  | GFG | clay | 12 |  | 2007-2009 |
| **regional** | REGIONAL | USDA texture maps (Fig. 2) |  | CRU-NCEPv7.1 (6 hourly) | 1981-2016 |

**2.2 Observations used as benchmarks**

**2.2.1 Site data**

At all three tropical forests sites (Bonal et al., 2008; da Rocha et al., 2009), measurements include hourly turbulent sensible

(H) and latent (LE) heat fluxes, and net ecosystem carbon exchange (NEE) made using the eddy-covariance technique (Baldocchi et al., 2001; Shuttleworth et al., 1984). Gross primary productivity (GPP) and ecosystem respiration were retrieved from NEE using the algorithm of *Reichstein et al.*, [2005]. Flux data are noisy, and *Hollinger and Richardson*, [2005] evaluated the relative uncertainty of H, LE and $CO_2$ fluxes derived from eddy-covariance measurements to be around 25% for a temperate site. For eddy-covariance data, energy balance closure is a good proxy for data quality (Wilson, 2002).

We therefore calculated the overall energy balance ratio as the ratio of the sum of outgoing radiation (LE + H) divided by the sum of incoming radiation averaged over the study period (Wilson, 2002). K67 and GFG showed a consistent energy closure (ratio of 1.008 and 0.96 respectively), but at M34 energy fluxes should be interpreted carefully as energy closure was not achieved (ratio of 0.69).

Reported LAI, basal area (BA), and canopy height, (references in Table 2), were used to benchmark the site-level simulations. At K67, a vertical soil moisture profile was available (Nepstad et al., 2007). At GFG, old-growth forest plots were surveyed (Gourlet-Fleury et al., 2004; Ho Tong Minh et al., 2016). We used tree diameter and height measurements (for 1592 trees) from the 2014 inventory on a 6.25 ha plot. Forest inventories used in this study only included trees measured





above a DBH of 10 cm. Data from a site near to GFG that had been clear-cut in 1976 and then left to regenerate were also used to evaluate forest regeneration in CAN and CAN-RS [Chave et al., in prep].

### 2.2.2 Regional datasets

5 To evaluate GPP patterns and seasonality at regional scale, we used the monthly global observation-based GPP model FLUXCOM, running from 1981 to 2013 and produced at 0.5° resolution using different methods by *Tramontana et al.*, [2016] and *Jung et al.*, [2017]. We calculated the median of the following three methods, namely ANNs (artificial neural networks), RF (Random Forest) and MTE (Model Tree Ensemble) and chose the method proposed by *Lasslop et al.*, [2010] to retrieve GPP by fitting a respiration model to nighttime NEE values. All methods were highly consistent (not shown).

10 Compared to the global network of flux-tower measurements, performances were reasonable in terms of annual mean and spatial pattern representation ($R^2 > 0.7$) and mean seasonal cycle ($0.67 < R^2 < 0.77$), but they showed a low predictive power for interannual variability (Tramontana et al., 2016). Also, GPP (and other fluxes) were better predicted in temperate climate sites than in the tropics due to a smaller amount of training data being available (Tramontana et al., 2016).

15 For ET, we used the remotely sensed GLEAM v3.1a product [*Martens et al.*, 2017 and references within] interpolated at 1-degree resolution from 1981 to 2016. To illustrate the uncertainties associated with this global dataset, GLEAMv3.1 was compared to the ET measurement at K67 (M34) between 2000 and 2006, *Moreira et al.*, [2018] found a relatively strong bias of 0.77 (0.99) mm d$^{-1}$ and low correlation -0.08 (0.32).

20 Furthermore, we used a compilation of 413 ground inventories across Amazonia presented by *Mitchard et al.*, [2014]. Basal area (BA) was directly calculated from diameter measurements, and aboveground biomass (AGB) was retrieved using the three-parameter moist tropical forest allometric model of *Chave et al.*, [2005].

## 3. Results

### 3.1 Site-level evaluation of the models

While TRUNK has been evaluated over Amazonia (Getirana et al., 2014; Guimberteau et al., 2012, 2014), CAN has been evaluated for European forests only. All three model versions (TRUNK, CAN and CAN-RS) predicted the yearly mean state

30 of forest features (such as LE, GPP, LAI) with a bias < 20% at the three test sites (Table 2). CAN and CAN-RS predicted more productive forests with higher biomass (higher LE, GPP, AGB, LAI) than TRUNK. At M34, CAN and CAN-RS overestimated LE, but the energy budget was not closed at this site (see Methods). TRUNK simulated an AGB lower than the observed one, and CAN and CAN-RS generally overestimated AGB, especially at K67, where the overestimate by CAN and CAN-RS was 25% and 31%, respectively. This overestimation of AGB may possibly result from recent disturbances





[Pyle et al., 2008] that the models did not take into account. Finally, CAN and CAN-RS tended to underestimate tree height

and overestimate basal area (Table 2).

**Table 2. Comparison of TRUNK, CAN and CAN-RS against observations made at K67, M34 and GFG. Mean percentage bias between the observations and model results are highlighted in green when < 20%, in blue when between 20 to 40% and in red when > 40%.**

| VARIABLE | Site | OBS | TRUNK | CAN | CAN-RS | Refs and remarks |
|---|---|---|---|---|---|---|
| **LE** (W m$^{-2}$) | K67 | 86 | 70 | 87 | 89 | Eddy-covariance measurements (Bonal et al., 2008; da Rocha et al., 2009) |
| | M34 | 79 | 91 | 132 | 131 | |
| | GFG | 119 | 92 | 102 | 115 | |
| **GPP** ($\mu$mol CO$_2$ m$^2$ s$^{-1}$) | K67 | 8.2 | 7.7 | 6.3 | 7.4 | |
| | M34 | 7.9 | 7.5 | 8.1 | 8.1 | |
| | GFG | 9.7 | 7.6 | 7.8 | 9.6 | |
| **AGB** (tC ha$^{-1}$) | K67 | 148 ± 3 | 101 | 198 | 214 | (Pyle et al., 2008) recently disturbed plot |
| | M34 | 180 ± 10 | 99 | 221 | 221 | (Malhi et al., 2009b) |
| | GFG | 203 | 102 | 206 | 228 | (Dubois-Fernandez et al., 2012) |
| **LAI** (m$^2$ m$^{-2}$) | K67 | 6.4 ± 0.1 | 4.9 | 6.2 | 6.6 | (Malhi et al., 2009a) |
| | M34 | 5.6 ± 0.2 | 4.8 | 6.7 | 6.7 | (Malhi et al., 2009a) (hemiphoto method) |
| | GFG | 8.6 ± 0.7 | 5.0 | 6.5 | 7.3 | *[Granier et al., 1996]* (Demon Leaf Area Method) |
| **Canopy Height** (m) | K67 | 29.1 ± 7.2 | - | 19 | 19.1 | (Meyer et al., 2018) (Fig. S3) mean canopy height model (CHM) at 1 m resolution from LIDAR and associated standard deviation |
| | M34 | 26.7 ± 6.8 | - | 19.5 | 19.6 | |
| | GFG | 29.7 ± 9.5 | - | 19.4 | 20.2 | |
| **Basal Area** (m$^2$ ha$^{-1}$) | K67 | 31 | - | 36.1 | 36.2 | *[Hunter et al., 2008]* |
| | M34 | 27 | - | 36.6 | 36.6 | *[Rodrigues et al., 2001]* |
| | GFG | 31.6 | - | 36.5 | 37.3 | *[Ferry et al., 2006] (Table 4)* |

### 3.1.1 Seasonal water and carbon fluxes

Looking at the time series of LE, GPP and NEE at the three sites, the three model versions displayed reasonable scores with temporal correlations between observations and simulations varying from 0.6 to 0.8; the normalized standard deviation and RMSE ranked from 0.5 to 1 (Fig. 3). CAN-RS represented the standard deviation better than TRUNK, but not the correlation. CAN-RS outperformed CAN at two of the sites, K67 and GFG, but not at M34 where the two models had a similar performance (Fig. 3).



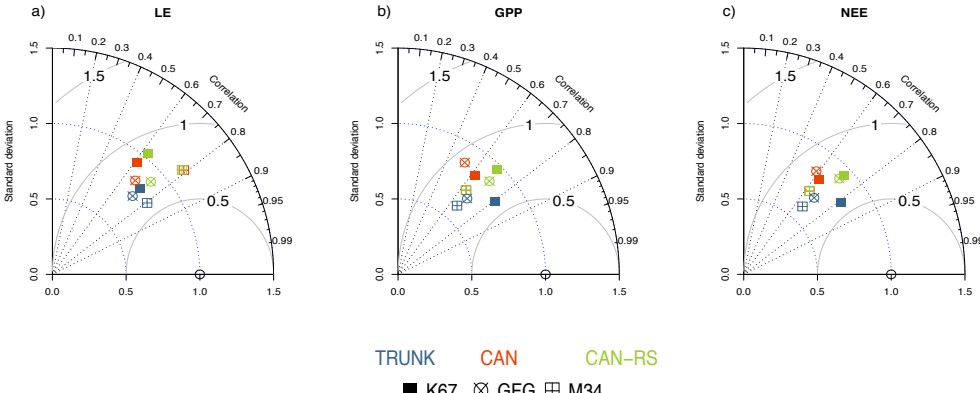

**Figure 3. Taylor Diagrams (Taylor, 2001) for: (a) LE, (b) GPP and (c) NEE, at three Amazonian sites equipped with a eddy-flux tower systems. These quantities were calculated among hourly values removing nighttime values (defined**
**by downwelling shortwave radiation ≤ 5 W m⁻²). In a Taylor diagram, correlation extends radially from the origin. The blue concentric lines indicate identical ratios of standard deviation of the simulated flux to the observed flux. The grey lines represent identical root mean square errors (RMSE) of the centred fluxes.**

The effect of the soil-to-root resistance-weighting scheme on LE and GPP was strongly influenced by the soil type (Table 1
and Fig. 2). Little difference between CAN and CAN-RS was observed at site M34, because there, the soil is loamy, implying a low water stress most of the year. At site GFG however, soil is clayey implying a high water stress during the dry season, and CAN-RS performed better than CAN, which underestimated LE and GPP by about 31% and 54%, respectively, during the dry season. At that site, TRUNK was also found to overestimate the seasonality of the fluxes. For site K67 with a clayey-loamy soil, implying an intermediate water stress, CAN-RS buffered the dry season drop in LE and GPP simulated by
CAN during the first (2002) and third years (2004). In 2003, CAN-RS simulated a decrease in LE and GPP two months sooner than CAN (Figs. 4 and 5).

CAN-RS better simulated the flux seasonality compared to CAN for soils prone to water stress during the dry season, by buffering the effect of drought stress. CAN-RS performance was comparable to TRUNK with, at daily time-steps, a better
representation of the variability for all fluxes, but lower correlations for carbon fluxes (Fig. 3). CAN-RS simulated a midday depression for GPP during dry seasons, which is not apparent in the data, resulting in a lower correlation between observations and simulations than for the TRUNK version (Fig. 4).





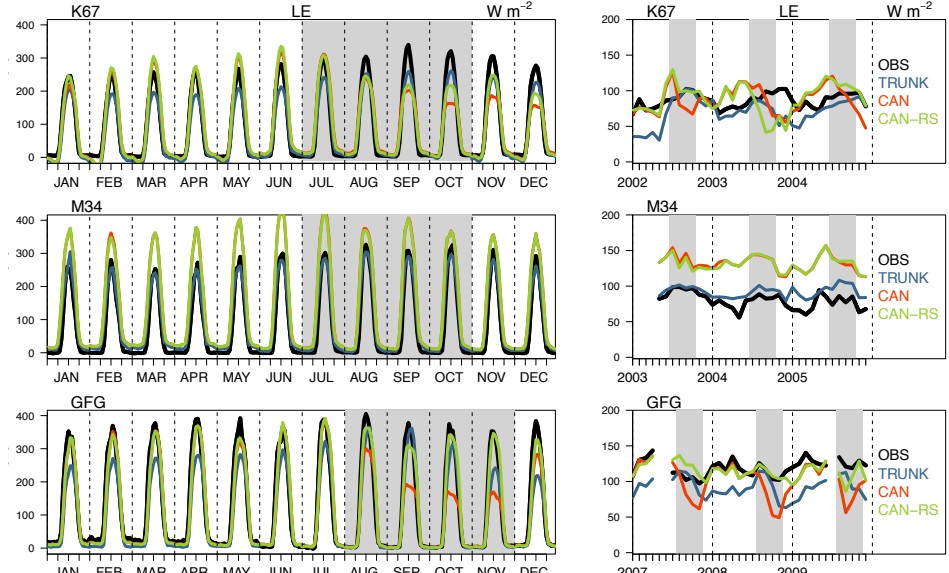

**Figure 4. Observed and simulated LE (W m$^{-2}$) at the three sites. Left panels show the average diurnal cycle for each month over three years; and right panels, monthly mean time series. Grey shaded areas indicate dry seasons (here defined as periods with precipitation less than 100 mm per month).**





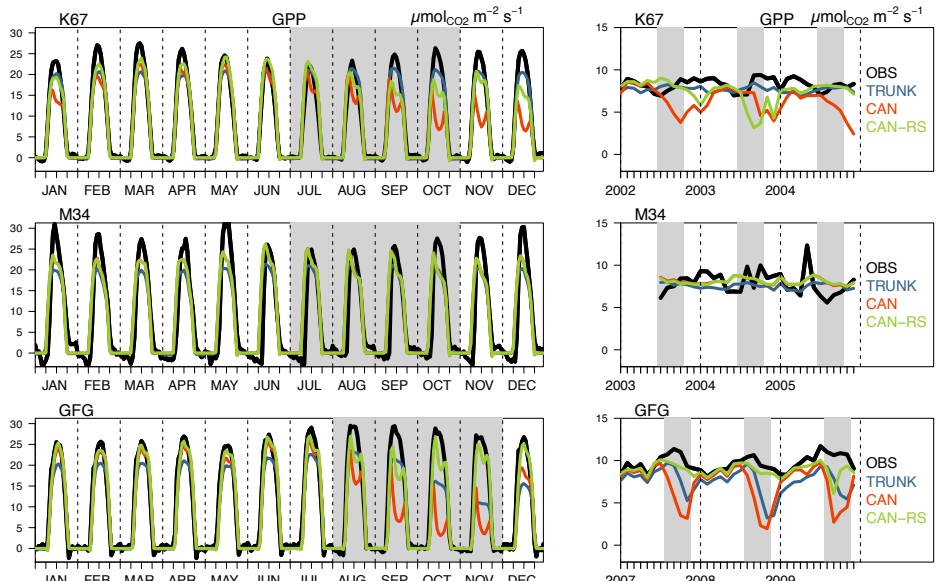

**Figure 5. Observed and simulated GPP ($\mu$molCO$_2$ m$^{-2}$ s$^{-1}$) at the three sites. Left panels show the average composite monthly diurnal cycle for each month over 3 years; and right panels, monthly mean time series. Grey shaded areas indicate dry seasons (here defined as periods with precipitation less than 100 mm per month).**

### 3.1.2 Soil volumetric water content and transpiration

To better understand the effect of the new root water uptake scheme, we focus here on the K67 site, where direct observations of the variation of soil water content with depth are available. Deviations between observations and simulations may be due to using soil texture and van Genuchten parameters from the USDA soil parameterization — these might deviate from actual soil at K67 (Figs. 6b-d). Besides, soil water content tended to be lower in CAN-RS than in CAN, especially during the dry seasons, in agreement with a higher LE simulated by CAN-RS (Fig. 4).

For the years 2002 and 2004, the soil-to-root resistance scheme implemented in the multilayer soil model allowed CAN-RS to overcome the too strong tree water stress simulated by CAN during dry seasons (Figs. 4 and 5) and $\Psi_{rz}$ stayed close to zero (Fig. 6e). Wet season rainfall restocked soil layers with water from top to bottom (Fig. 6d), and most layers then contributed to the transpiration flux (Fig. 6f). As the dry season progressed, the topsoil layers became drier due to stronger evaporation and harsher root competition, which induced a shift of water uptake towards deeper and wetter soil layers (Fig. 6f), where the soil-to-root resistance was lower (Eq. 6) and $E_{max}$ higher (Eq. 7). Since the 2003 wet season was drier (1276 mm) than the ones in 2002 (1683 mm) and 2004 (1849 mm) (Fig. 6a), the amount of precipitation was insufficient to



recharge the soil after dry-season depletion (Fig. 6b-c). This translates into strong hydrological stress during the 2003 dry season, with daily $\Psi_{rz}$ reaching -2.3 MPa (Fig. 6e). This failure to completely recharge the soil profile caused a significant reduction in the simulated LE and GPP in 2003 (Figs. 4 and 5).

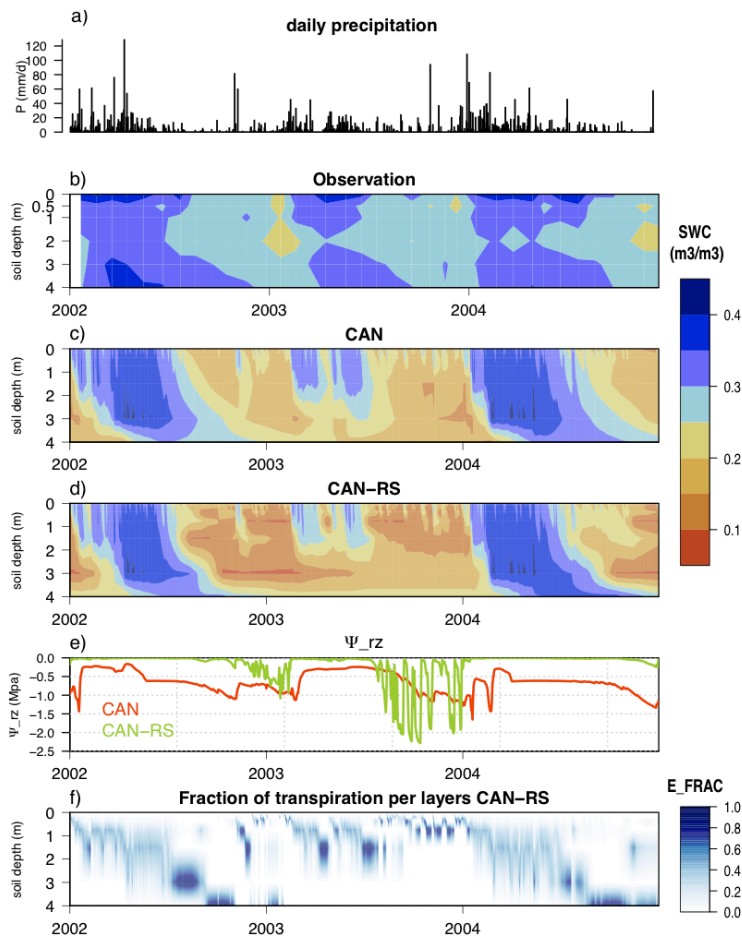

**Figure 6. Daily times series from 2002 to 2004 at K67 of (a) precipitation, (b) observed soil moisture profile, (c) soil moisture (SWC) profile simulated in CAN (d) and soil moisture (SWC) profile simulated in CAN-RS, (e) soil water potential in the rooting zone ($\Psi_{rz}$), and (f) simulated soil profile of the contribution of each layer to total root water uptake $E_{frac}(l)$ from CAN-RS, defined as $E_{max}(l)$ divided by the sum of $E_{max}$ across all layers.**





### 3.1.3 Forest structure

We found that both CAN and CAN-RS both correctly reproduced forest establishment from bare soil based on empirical data on forest regeneration (Chave et al., in prep) in French Guiana near GFG (Fig. 7), starting with a fast increase in AGB and

5   BA, which levelled off as self-thinning began.

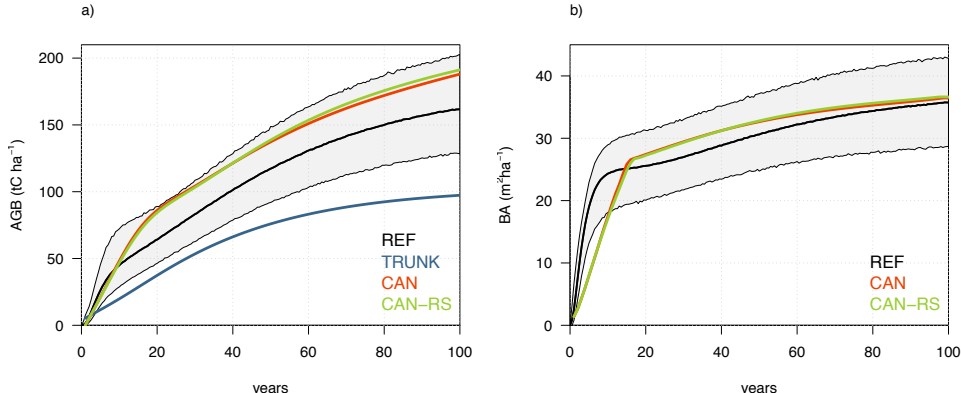

**Figure 7. Dynamics of (a) the aboveground biomass (AGB) and (b) basal area simulated by the different versions of ORCHIDEE soil during the first hundred years after clear-cut, compared to tree inventory data in the regeneration experiment ARBOCEL (REF) (Chave et al., in prep).**

The representation of the forest by CAN-RS (or CAN), while more realistic than a big-leaf model (such as TRUNK), remains an approximation because it considers only 20 classes of tree diameter and mono-specific (single PFT) parameters within the forest. When comparing the simulated and measured forest structure at GFG using a forest inventory and measured tree heights (Fig. 8), CAN-RS and CAN showed a realistic diameter-height allometric relationship (Fig. 8a) and a

15   diameter-size distribution with many small trees, and few large trees (Fig. 8b). However, the number of individuals was slightly overestimated in CAN-RS with 800 trees ha$^{-1}$ compared to 600 trees ha$^{-1}$ from the forest inventory.





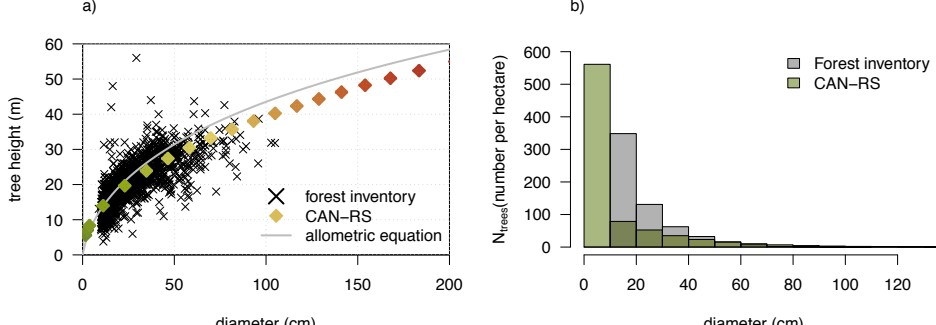

**Figure 8. Forest structure modelled in CAN-RS compared to forest inventory data over non-disturbed plots at Paracou (French Guiana), with (a) allometric relationship between tree diameter and tree height for the 20 simulated diameter classes in CAN-RS plotted in colours compared to 1592 measurements; plotted in grey, the diameter-height**
**allometric equation for tropical forest proposed by *Chave et al.*, [2014]; Eq. (6a); (b) mean diameter distribution per hectare for CAN-RS compared to the 2014 forest inventory of a 6.25 ha plot in Paracou.**

In CAN simulations we found more trees (930 trees ha$^{-1}$), but with a smaller mean diameter (Fig. 9a). The higher GPP in CAN-RS than in CAN (especially during the dry season; Fig. 4) allows CAN-RS to grow more large trees, (Fig. 9a), leading

to a higher self-thinning effect, and slightly fewer saplings and poles than in CAN (Fig. 9b). This difference in forest structure translates into a higher AGB than in CAN (228 versus 206 tC ha$^{-1}$, Table 2).

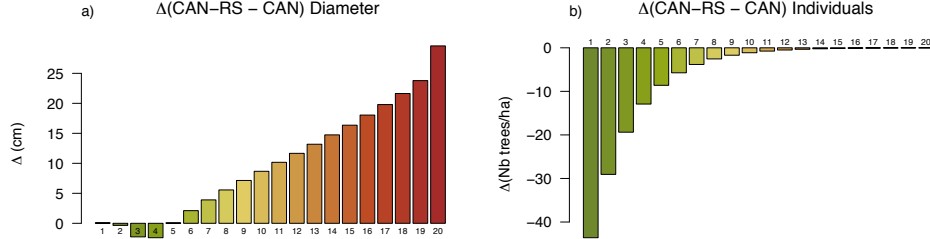

**Figure 9. Comparison between CAN-RS and CAN for: (a) mean trunk diameter per cohort, and (b) number of trees**
**per cohort. Cohorts are illustrated as coloured bars numbered from 1 to 20, at Paracou, French Guiana (GFG).**

**3.2 Regional evaluation**

**3.2.1 Carbon and water fluxes**





Compared to the in situ fluxes at the three sites, both TRUNK and CAN-RS simulated ET and GPP reasonably well, except at M34 (Fig. 4 and Table 2). At regional scale, however, both models slightly overestimated annual LE (Fig. 10) and GPP (Fig. 11) when compared to the regional GLEAM LE and FLUXCOM GPP products. Differences can be partly explained by the fact that at local scale, models were forced with hourly local meteorological data, while for regional simulations we used

the 6-hourly CRU-NCEP fields. Another explanation is the large uncertainties associated with the regional products over the tropics. For example, both models correctly simulated total ET as compared with the ET product from *Jung et al.*, [2011] covering the 1982-2008 period (Fig. S1).

ET and GPP simulated by CAN-RS reproduced the spatial pattern from GLEAM ET and FLUXCOM GPP, with higher

annual fluxes in the northeast and southwest of Amazonia, and lower GPP along the southeast border (Figs. 10a;c, 11 a;c). In CAN-RS these patterns were mainly driven by a relatively high downwelling shortwave radiation, and higher precipitation in these areas (Fig. S2). TRUNK simulated a more homogeneous pattern and was less sensitive to climate gradients than CAN-RS.

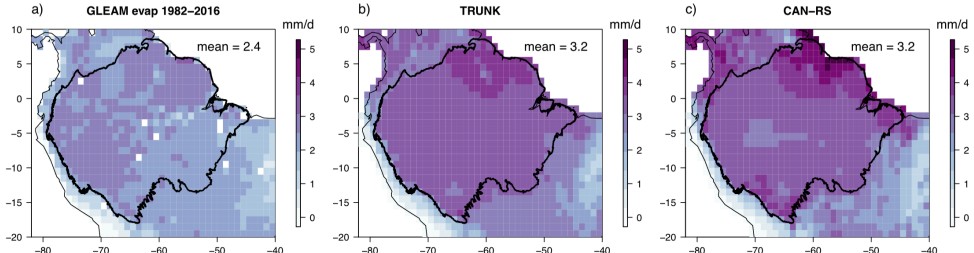

**Figure 10. Annual mean (1982-2008) ET simulated by (b) TRUNK and (c) CAN-RS compared to the GLEAM product (a).**

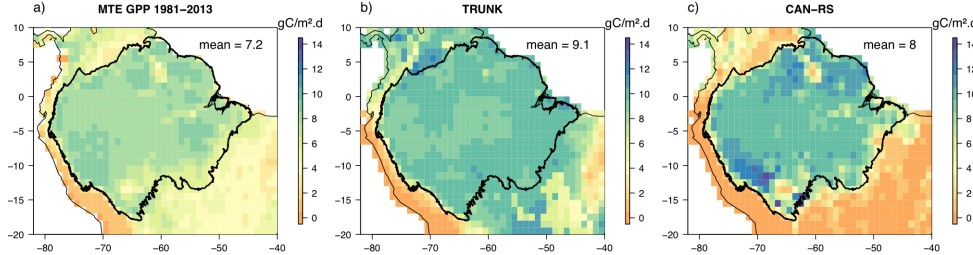

**Figure 11. Annual mean (1981-2013) GPP simulated by the TRUNK (b) and CAN-RS (c) compared to the**
**FLUXCOM GPP product (a).**



CAN-RS simulated higher annual mean ET and GPP than CAN over the Guianan and Brazilian Shields (Fig. 2, Fig. 12a, Fig. 13a). Comparison of monthly time series averages across the shields (Figs. 12a and 13a) shows that CAN-RS gives a better fit than CAN for both ET (0.81 versus 0.21 in the Guianan Shield, and 0.52 versus 0.40 in the Brazilian Shield) and GPP (0.42 versus 0.32 in the Guianan Shield, and 0.73 versus 0.67 in the Brazilian Shield). CAN simulated a drastic

reduction in LE and GPP during the dry seasons (Fig. S3) and even in the months following, while CAN-RS simulated a higher dry-season ET and GPP that better matched the dry-season observations (Figs. 12b and c, Figs. 13b and c) and simulated a more realistic SWC-GPP relationship (Fig. 14).

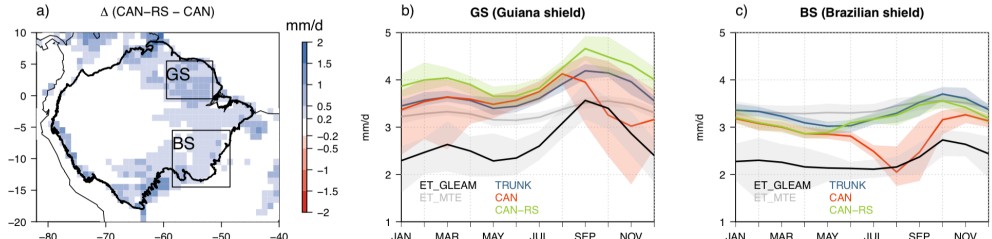

**Figure 12. (a) Difference in predicted annual mean evapotranspiration (ET) between the**
**simulations of CAN-RS and CAN from 1982 to 2016. Comparison of the reference (GLEAM) and simulated ET (TRUNK, CAN and CAN-RS) mean seasonal cycle over (b) the Guianan Shield, and (c) the Brazilian Shield, including all pixels with at least 50% cover by evergreen tropical forest. The envelopes represent the monthly minimum and maximum over the entire period for each variable.**

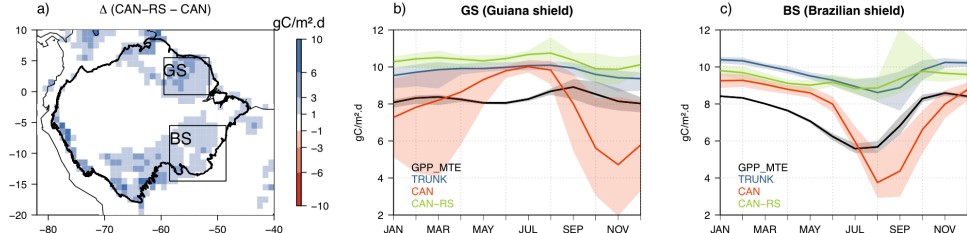

**Figure 13. (a) Difference in predicted annual mean GPP between the simulations of CAN-RS and CAN from 1981 to 2013. Comparison of the observed GPP and simulated GPP (TRUNK, CAN and CAN-RS) mean seasonal cycle over (b) the Guianan Shield, and (c) the Brazilian Shield.**
**Envelopes describe the minimum and maximum values over the considered period, including all pixels with at least 50% cover by evergreen tropical forest.**





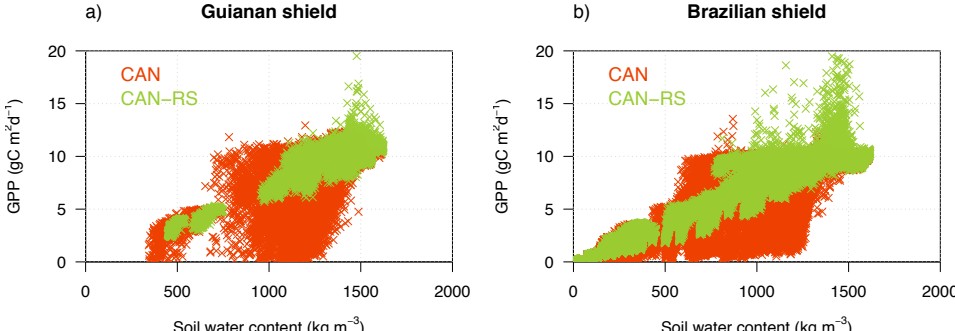

**Figure 14. Scatter plot of monthly GPP vs. monthly soil water content over (a) the Guiana Shield and (b) the Brazilian Shield simulated by CAN and CAN-RS from 1982 to 2016**

5 **3.2.2 Carbon stocks**

The AGB stocks simulated by TRUNK were half of those simulated by CAN-RS (Figs. 15a and b), and CAN-RS also predicted a higher AGB than CAN over the Guianian and Brazilian Shields, due to higher GPP during the dry seasons (Fig. 13). This is explained by differences in parameterization and in the tree mortality processes, keeping in mind that only 10 crowding mortality is modelled and not climate-induced (drought) mortality. In Amazonia, the simulated carbon-use efficiency (CUE, the ratio of NPP to GPP) was higher in CAN-RS (0.42) than in TRUNK (0.30), because of lower maintenance respiration ($R_{A,m}$) in CAN-RS (3.1 versus 5.3 gC m$^2$ d$^{-1}$) as the growth respiration was slightly higher (1.4 versus 1.1 gC m$^2$ d$^{-1}$). CAN-RS overestimated AGB, especially in the southwest part of the basin, but we emphasize that the model was calibrated using sites with relatively high AGB mainly located in the northeast of Amazonia (Table 1, Fig. 2 and 15 Fig. 16). Results from both models fit in the range of CUE field observations ranking from 0.27 to 0.52 (Malhi et al., 2009a, 2015), but in CAN-RS (and CAN), $R_{A,m}$ is calculated for each living compartment as a function of temperature, biomass, prescribed carbon/nitrogen ratio and $k_{cmaint}$, the fraction of allocatable photosynthates consumed for maintenance and growth respiration (which is a tunable parameter, see Table S2). Parameter $k_{cmaint}$ is poorly constrained by observations (Sitch et al., 2003), and it can be optimized (e.g., as by Naudts et al. (2015)) for each site) or manually tuned, as in this study. TRUNK 20 simulated a lower CUE, but underestimated AGB (Table 1, Fig. 15). Contrary to CAN in which mortality is an emerging result of modelled competition processes via self-thinning (Fig. 1), in TRUNK the background tree mortality is very simply set as a constant fraction of the woody carbon pool, defined by a "residence time" parameter which is poorly constrained by observations (Sitch et al., 2003). Thus, it is possible to capture a realistic AGB with TRUNK by adjusting this residence time value, while in CAN-RS and CAN tuning the model to reproduce AGB is not trivial. Nevertheless, the three model versions




appear to lack the mechanisms underlying the observed SW-NE gradient in AGB, that have been hypothesized to relate to soil fertility (phosphorus), tree species or soil mechanical resistance.

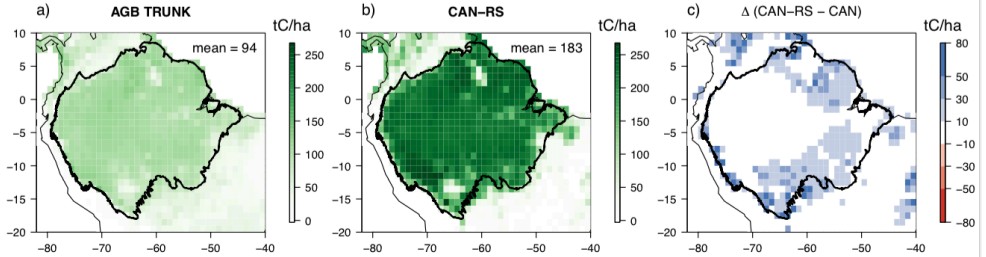

**Figure 15. Annual mean of aboveground biomass (AGB) from 2000 to 2010 simulated by (a) TRUNK and (b) CAN-RS. Panel (c) shows the difference between CAN-RS and CAN.**

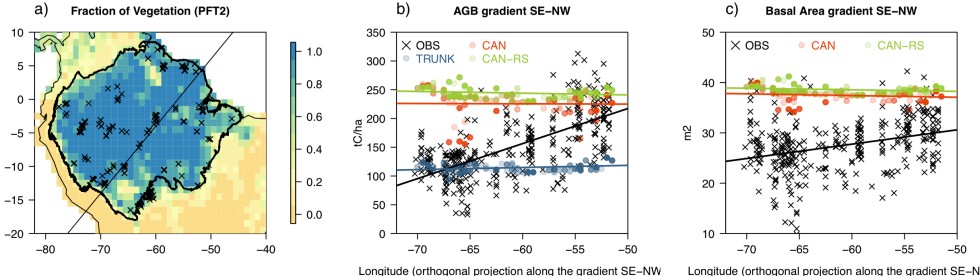

**Figure 16. (a) fraction of evergreen tropical forests input in ORCHIDEE and location of the in situ inventories collected by *Mitchard et al.*, [2014]; (b) comparison of simulated aboveground biomass and (c) basal area from the field inventory measurements along the SE-NW gradient indicated in (a).**

## 4. Discussion

### 4.1 Root water uptake module and soil hydraulic parameters

With the mechanistic root water uptake module implemented in CAN-RS version of ORCHIDEE, trees preferentially used water in the deepest soil layer during the dry season. This reduced the effective decrease in leaf gas exchange induced by water stress. Through this process, the model better captured the seasonality of GPP and LE. The effect was strongest for soil types with a constraining soil water retention curve (e.g., clay USDA soil type 12). The root water uptake model was successfully validated under severe drought conditions (Fisher et al., 2006) and for other ecosystems (e.g. by *Williams et al.* (2001)), and can therefore be extrapolated to other PFTs for global simulations using CAN-RS. CAN, like other DGVMs



(Getirana et al., 2014; Guimberteau et al., 2014; Restrepo-Coupe et al., 2016), was unable to maintain LE and GPP fluxes during the dry season.

A previous modelling study using TRUNK showed the importance of deep roots by optimizing soil depth parameters at 10 m
for seasonal variation of fluxes in Amazonia (Poulter et al., 2009; Verbeeck et al., 2011). However, this is a model artefact as TRUNK's water stress function directly links available soil moisture content to leaf gas exchange. Increasing soil depth in TRUNK will always increase the water storage capacity, but will not shift tree water uptake to the deepest layers during dry seasons since the root profile is fixed and exponentially decreases from top to bottom in the soil column. By using the soil-to-root resistance weighting scheme and a relatively deep soil (4 m) that contains most of the fine roots (as observed by
Markewitz et al. (2010); Nepstad et al. (1994); Schenk and Jackson (2005)), CAN-RS sustains productivity by explicitly simulating the observed shift of water uptake to deeper and wetter soil layers during the dry season (Moreira et al., 2000).

Because of its explicit hydraulic architecture, CAN and CAN-RS are more sensitive than TRUNK (water stress function) to the parameters of the Mualem-van Genuchten Model (van Genuchten, 1980; Mualem, 1976). It is well known that changes
in the spatial resolution of the soil input data by aggregating small-scale information causes serious problems in models (Van Looy et al., 2017), as well as the use of coarse soil texture classes (Kishné et al., 2017). Thus, along with improving model representation of the hydraulic gradient from the soil to the plant in DGVMs [this study, *Sperry et al.*, 2002; *Fisher et al.*, 2006], it is important to improve the parameterization of the physical soil environment (Marthews et al., 2014). While increasing the spatial resolution is computationally expensive, accounting for variability of soil parameters rather than
combining these parameters into texture classes (e.g., USDA texture class) may now be explored using, for example, hydraulic parameters maps as developed by *Marthews et al.*, [2014] at regional scale.

**4.2 Missing processes to correctly capture spatial variability across Amazonia**

As with most DGVMs (Castanho et al., 2015; Johnson et al., 2016), TRUNK, CAN and CAN-RS fail to capture the SW-NE gradient of AGB (and BA) across Amazonia (Fig. 16) — they simulate a quasi-constant AGB across the basin (Fig. 15). This AGB gradient could be caused either by productivity or tree mortality. Spatial variation in wood productivity can be linked to spatial variability in soil properties [*Quesada et al.*, 2012], like soil fertility [*ter Steege et al.*, 2006 ; *Malhi et al.*, 2004, Turner et al., 2018] and soil hydraulic parameters. Therefore, the incorporation of detailed soil hydraulic parameters maps
(e.g., [*Marthews et al.*, 2014]) and inclusion of nutrient cycles into ORCHIDEE would yield advances.

Besides variation in productivity, it has been shown that variation in tree mortality is a key driver of AGB across Amazonia [*Johnson et al.*, 2016 and references within]. TRUNK, CAN and CAN-RS show a quasi-linear positive relationship between NPP and AGB, despite their differences in forest representation and tree mortality scheme over Amazonia (Fig. 18a) and





when focusing on different regions of the basin (Figs. S4 and S5). For high NPP, CAN-RS and CAN showed a saturation of GPP (not shown) and AGB around 250 tC ha$^{-1}$ (Fig. 17a) due to a breakpoint in precipitation at 2000 mm yr$^{-1}$ (Fig. 17b), this phenomenon was also identified by *Ahlström et al.* [2017].

CAN-RS and CAN still lack functional diversity as a calibration of the self-thinning curve (Fig. 1), and all parameters including wood density are constant within the "evergreen tropical forest" PFT covering most of Amazonia (Fig. 16a). Therefore, having a size distribution representation of the forest, as suggested by *Johnson et al.*, [2016] is not sufficient, functional diversity needs to be accounted for in 2gDGVMs.

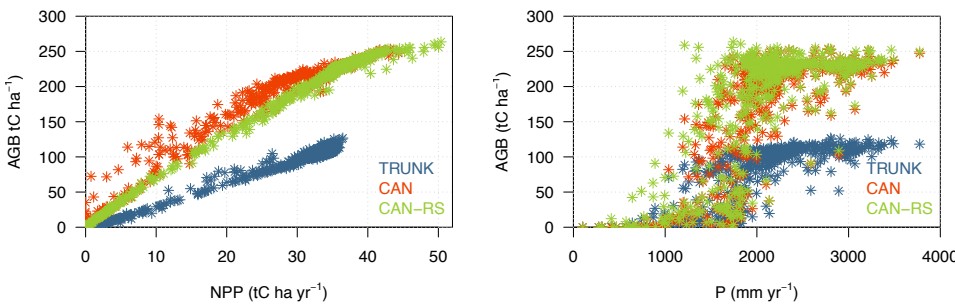

**Figure 17. (a) scatter plots of mean AGB from 1981 to 2016 plotted against mean annual NPP and (b) annual precipitation averaged over the same time period for Amazonia.**

**4.3. Concluding remarks**

In this study, we demonstrated that a 2gDGVM (CAN-RS) can reproduce carbon and water fluxes and carbon stocks across Amazonia, and we validated these simulations using local and regional observations. We also compared CAN-RS with earlier versions of the same family of models (ORCHIDEE-TRUNK and ORCHIDEE-CAN). The mechanistic root water uptake module implemented in CAN-RS allows trees to take up water in the deepest soil layer during dry seasons. This feature enables the model to capture the seasonality of GPP and LE, especially over the Guianan and Brazilian Shield (high

water retention). The modelled diameter size distribution is improved compared to the TRUNK big-leaf DGVM. However CAN-RS, by construction, still does not capture functional diversity, and behaves similarly to TRUNK. To include functional diversity in a 2gDVM, especially for tropical ecosystems, we believe that using soil hydraulic parameter maps (e.g., those from *Marthews et al.*, [2014]) as an input is a necessary first step. Second, and because of the negative relationship between soil fertility and wood density (Baker et al., 2004; Patiño et al., 2009; ter Steege et al., 2006), and wood



density and tree mortality (King et al., 2006); wood density should vary across the basin (rather than using a single parameter for the evergreen tropical forest PFT); this requires the use of wood density maps (ter Steege et al., 2006). Finally, mortality processes need to be linked to edaphic properties. Accounting for functional diversity in a 2gDGVM is crucial, not only to capture spatial patterns of Amazonian forest, but also to estimate its resilience to climate change (Sakschewski et al., 2016).

**Aknowledgments**

Data acquisition in French Guiana was supported by an "investissement d'avenir" grant from the Agence Nationale de la Recherche (CEBA, ref ANR-10-LABX-25-01). J.B. acknowledges support from (CR)[2] Chile (CONICYT/FONDAP/15110009). Matthieu Guimberteau, D. Goll and P. Ciais are funded by the European Research

Council Synergy grant ERC-2013-SyG-610028 IMBALANCE-P. We also acknowledge the European Union Climate KIC grant FOREST Specific Grant Agreement EIT/CLIMATE KIC/SGA2016/1CNES (TOSCA program) for funding.

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

**Code availability**

The code of ORCHIDEE-CAN r2290 (Naudts et al., 2015) can be accessed from http://dx.doi.org/10.14768/06337394-73A9-407C-9997-0E380DAC5595