# Peer review of "The importance of tree demography and root water uptake for modelling the carbon and water cycles of Amazonia"

_Biogeosciences, 2018_

## Referee Comment (RC1) · Anonymous Referee #1 · 26 Sep 2018

The manuscript by Joetzjer et al. explores how two new formulas for the ORCHIDEE dynamic global vegetation model influences carbon and water budget predictions of three Amazon forests. ORCHIDEE-DGVM, called TRUNK in this manuscript, is used as the baseline model and represents the land surface physiology as an aggregated 'big leaf'. The first new formula, ORCHIDEE-CAN (CAN), represents tree demography and aboveground mechanistic water transport through the stem, and is parameterized specifically for Amazon forests. The second new formula (CAN-RS) modifies CAN to include mechanistic soil water uptake by the roots. Improving the representation of belowground processes in land surface models is at the forefront of discussions in the modelling community and is a recommendation in the recent paper by Fisher et al.

[Figure]

(2018), which is an authoritative review paper about the state-of-the-science of second generation dynamic vegetation models. I read this manuscript with great enthusiasm since these two formulas, CAN and CAN-RS, are at the cutting edge of model development needs. I do feel that this manuscript has great potential to add to our scientific knowledge about individual tree water-use strategies and physiology. However, in my opinion this manuscript is still too premature to be considered for publication. I support my opinion with the following comments.

General comments.

1. My main critique of this manuscript, and one that I feel should be sufficiently addressed before it is considered for publication, is that it presents three alternative hypotheses about biological controls over tropical forest carbon and water cycles, yet these hypotheses are not clearly set up in the Introduction, described in the Methods, or convincingly interpreted in the Results and Discussion (comments on each section are detailed below). Also, given the results, the manuscript does not convincingly justify the asserted need to increase the TRUNK model complexity specifically with the new CAN-RS formulation.

Introduction.

(a) I recommend eliminating or reducing the first paragraph to a single sentence in order to create more space to describe the three hypotheses being evaluated.

(b) It is not clear what hypothesis TRUNK represents that is relevant to this study. I presume it has something to do with spatial aggregation; but since there is only a single line (P3, ln7) written about TRUNK, it is unclear what CAN and CAN-RS are being compared to. It would be helpful to have some context about where TRUNK has been a good hypothesis for explaining forest function, and carbon and water exchanges, and in what cases the hypothesis is not well supported.

(c) I feel that the Introduction will flow better if the three hypotheses that are all jointly

introduced at the end of the Introduction are instead introduced in each's paragraph that contains its respective problem statement. For example, in paragraph three of the Introduction, (P2, ln23-29), it would be clearer if the proposed remedy (i.e. increased "layer-to-layer heterogeneity in soil-to-root resistance) is included as the concluding sentence of this problem-statement paragraph.

(d) It would be helpful to articulate in the Introduction what "fingerprints" the data are expected to have that would support of each particular hypothesis. E.g. what fingerprint (or marker) should be in the SWC data to support the given dynamic-root-water-uptake hypothesis? (Conversely, if dynamic root-water uptake is not important, what should the SWC data look like?) Articulating the fingerprints gives a justification for the selection of each benchmark presented. As the paper is currently written, the reader is first introduced to the specific benchmarks in the Results and has to intuit the rationale for each.

Methods.

(a) Eqns. 3,4, 5 and 6 need to include an index (e.g. i) for each variable that is calculated separately for each soil layer (for example, if psi_s at P5 ln10 is calculated by layer, it should be psi_s,i, and i needs to be defined). Similarly, it is not clear how the model differentiates between the resource-use of different-sized trees. Should there be an index (e.g. j) in Eqn 5 for each size class of trees? Or, are all the roots uniformly distributed such that the seedlings have the same rooting depth and access to water as the canopy trees? Please clarify how this works. If it is the latter, then the new hydrodynamic physiology is not mechanistically linked to demography.

(b) I feel that the language related to the demography hypothesis needs to be much more precise. It is important to distinguish between whether the demographic rates emerge or are prescribed. If the demographic rates emerge due to a mechanism, then what is the hypothesized mechanism (C-starvation, hydraulic failure, etc.)? If the demographic rates are prescribed by the user, then this model does not simulate selfthinning through competition for limiting resources (P4 ln18) and it does not simulate demography as suggested in the Methods. Rather, the model represents the outcome of competition and self-thinning by prescribing a number-density by size-class distribution. In other words, does the model neglect all of the size-dependent internal dynamics of the ecosystem that give rise to the demographic rates from the bottom-up? If that is the hypothesis, then it needs to be clarified. Eqn 5 will act differently on individuals of different sizes due to non-linearities, than it will on all of the roots aggregated together. Making this distinction has very important implications for how ecologists and physiologist interpret the results of the simulations.

(c) The soil-water stress function for TRUNK needs to also be included. Is it similar to Eqn 3?

(d) I recommend that all parameters that are new or have new values introduced in this study be placed in a Table that includes the appropriate reference. It is a little cumbersome to have to search the text for this information.

Results and Discussion

(a) I am not convinced by the interpretation of the results for CAN and CAN-RS in terms of the claims that this paper is trying to make. The title implies that demography (CAN) and root water uptake (CAN-RS) are important for modelling tropical forest carbon and water cycles. The Abstract claims that modeling root water uptake in greater detail (CAN-RS) improves model performance (P1, ln32-35). The Results, Discussion and Conclusions make similar claims throughout. I do not feel that these conclusions are supported by the evidence presented in the Results for two primary reasons.

First, the Results need to give a much more thorough description of the Taylor Diagram (Figure 3); three sentences is not adequate (P10, ln8-12). Contrary to the Title and Abstract, the Taylor Diagram in Figure 3 indicates that TRUNK is an equivalent, but likely better, predictor of LE, GPP and NEE than CAN or CAN-RS. This is a highly significant outcome, yet it seems to be downplayed. In all three panels, TRUNK has

better or equivalent RMSE and correlation scores compared to the two other models. CAN and CAN-RS seem to only score better than TRUNK for the standard deviation metric. RMSE and correlation are indicators that the model is getting the pattern, and hence the mechanism, correct, whereas standard deviation indicates how well the mechanism is tuned to correctly capture the magnitude of the response. Therefore, for this type of analysis, which explores mechanistic controls, RMSE and correlation are more important indicators of model performance; yet the Results focus primarily on standard deviation. One objective way forward is to assign a skill score to each model as proposed by Taylor (2001) in Eqn 4 or 5. If a "penalty" is imposed, as in Eqn 5 of Taylor (2001), then the justification for the penalty needs to be described in the Methods.

Second, in just about all cases, TRUNK appears to be a better predictor of the benchmarks than either CAN or CAN-RS (Figures 3, 4, 5). To begin with, TRUNK seems to match the observations in Figure 4 the best of the three models. CAN seems to be the weakest predictor, which tells me that either demography itself is not important, or the hypothesis contained in CAN about how to represent demography is not supported. Which is it and why? Next, both CAN and CAN-RS show a midday suppression in GPP during the dry season that is not present in the observations (Fig. 5); however, TRUNK does not show the midday suppression. This tells me that the mechanistic water transport hypothesis as it is represented in CAN and CAN-RS is not supported by the observations. In Figure 6, neither CAN nor CAN-RS reproduce observed SWC in any credible way. I am curious about what fingerprint the authors would expect the data to have in support of the CAN-RS hypothesis (this should be mentioned in the Introduction). Does Fig.6b actually possess such a fingerprint? On the other hand, the observations presented in Fig. 5 do not possess the expected fingerprint (i.e. midday suppression). If the observations do not possess the expected fingerprints, then what is the rationale for including the CAN-RS hypothesis in TRUNK in the first place? And, what is the rationale for using the eddy flux and SWC data as benchmarks if they do not contain the relevant fingerprints? I do wonder if the observed SWC data presented

in Fig. 6 has the correct resolution to be a valid test for the CAN-RS hypothesis. If not, then the authors might consider Miller et al. (2011), it contains the fingerprint that supports the CAN-RS hypothesis (Fig. 1b) and perhaps would be a better test of the CAN-RS hypothesis. Finally, given that TRUNK has the correct pattern (and no midday suppression) but the incorrect magnitude in Figure 5, could it be that that the structure of TRUNK is correct, but it just needs better tuning?

(b) The Taylor Diagram is certainly a good method for cross-model comparisons to observations, but one of its limitations that should be addressed in the Discussion is that, unlike AIC for statistical models, it does not account for the trade-off between model simplicity and complexity. As acknowledged in the Introduction (P3, ln3), process-based land surface models have extremely high degrees of freedom that can lead to compensating errors and equifinality. It is unclear if and how this trade-off has been considered in the conclusion that CAN-RS is an improvement over TRUNK.

2. This manuscript would be much more impactful with greater synthesis with the many recent and important advances in the field of dynamic vegetation model development. The manuscript also needs greater synthesis about how these different hypotheses inform our physiological and ecological understanding of tropical forests.

(a) For example, Xu et al. (2016) and Powell et al (2018) also use a dynamic vegetation model that explicitly represents demography and mechanistic water movement through the soil-plant-atmosphere continuum. Xu. et al (2016) explores mechanisms related to water-stress avoidance, while Powell et al. (2018) explores mechanisms related to tolerance of water-stress. There is a good opportunity to connect the insights into belowground mechanisms explored in this study to the insights into aboveground mechanisms explored in those two studies.

(b) This study also addresses some of the issues raised in the recently published Anderegg et al. (2018).

(c) Figure 15. Figures of the spatial distribution of AGB across the Amazon basin have

been widely published. Zhang et al. (2015) reported such figures for several DGVMs (Fig. 3s) including one like CAN that contains a size-structure and demography hypothesis. I feel that Figure 15 is a significant result because the CAN hypothesis does not show a strong spatial gradient in AGB, which is in contrast to two hypotheses in Zhang et al. (2015) that do capture the gradient reasonably well (see Figure S3f in Zhang et al.). Why is the CAN hypothesis not supported by the data, but the other two hypotheses are supported by the data? Is it due to CAN model structure, experimental design, parameterizations? What can we learn about the ecology and model development needs from the contrasting results?

(d) Finally, this study and Levine et al (2016) proposed two different hypotheses about how to represent demographic processes in a land surface models; and, the two studies produces very important and contrasting results. Levine et al. (2016) argues that the size-structured model hypothesis is supported by the data because the demographic rates emerge from a bottom-up formulation of spatial heterogeneity. The CAN formulation, in contrast, appears to be a top-down scaling hypothesis. Given that the bottom-up approach agrees with benchmark tests (e.g. Xu et al. 2016, Zhang et al. 2015, Powell et al. 2018), but the top-down approach (CAN) does not perform better than TRUNK (a spatially aggregated hypothesis), what does this tell us about these approaches for representing demographic processes?

Making these connections will provide a much more complete picture about the contribution this study makes to the state-of-the-science regarding demographic dynamic vegetation models, modeling plant hydrodynamics, and understanding tropical forest ecology in general.

3. The total number of Figures could be reduced. The manuscript loses focus between Figures 10 and 17 and the rationale for these figures is not very well established in the Introduction (i.e. I do not see anything regarding predictions of spatial variability in the Introduction). I recommend this paper focus on figures that directly test the specific demography and root water uptake hypotheses contained in TRUNK, CAN,

and CAN-RS. Figures 10 to 17 are summaries and their relevance is predicated on the CAN and CAN-RS hypotheses being more strongly supported by the data than the TRUNK hypothesis. Also, the Discussion needs to be better integrated with the figures presented in the Results. All key results figures should be cited in the Discussion; if they are not, then this tells me that they are not central to the story of the manuscript and should be moved to the Supporting Material.

Specific comments.

P1. Line 1. Title. The title highlights the importance of tree demography, yet the experiment is not set up to explicitly isolate this, as CAN, contains at least three differences from TRUNK: (1) explicit tree demography, (2) mechanistic stem water transport, (3) Amazonian specific parameterization. With the CAN versus TRUNK comparison, the authors need to evaluate how each of these differences (as well as any others that may be unmentioned) individually impact model predictions. Otherwise, attribution to any particular modification is confounded by its interaction with the other two modifications.

P1. Line 22. "remains challenging". This is very vague and will mean many different things to many different people, much of which is unrelated to the subject of this manuscript. This needs to be clarified to keep the reader focused on the subject of this paper.

P1. Line 23. "These" lacks an antecedent and therefore it is unclear what the limitations are.

P1. Lines 25-28. Why evaluate these three hypotheses? The Abstract lacks a problem statement with sufficient detail to motivate their evaluation.

P1. Line 31. "as well as TRUNK...at local and regional scales..." If this is the case, then why use a more complex model? This statement also does not support the conclusion in line 34, "...improves the representation of biogeochemical cycles..."

P1. Line 35. Last sentence. This sentence is of course true...about a lot of things.

However, this sentence is quite vague and therefore loses any specific meaning with respect to the key findings of this study. It should be revised to specify "the variation [in what aspect of] ecological functioning". Plant hydraulics? Soil hydraulics? Life history strategies? Phenology? Biogeochemical cycling? Life forms—palms, shrubs, trees, lianas? Disturbance regimes? Etc.?

P2. Line 1. "will likely". This is too strong of a statement. Perhaps substitute " are predicted to".

P2. Line 8. The "large-scale dieback" is a quite dated result that has been updated widely in the literature over the subsequent decade and a half. Consider revising to be more current with our understanding of the system.

P2. Line 20. Xu et al. (2016) should be referenced here.

P5. Eqn 4. Is there a typo? Change min to max?

P10. Table 2. AGB. I think it is worth noting that none of the models capture the observed spatial variability of AGB.

P15. Line 16. Exchange "slightly" with "by 33%" (the two are probably not equivalent).

Figure 9a. y-axis. Delta of what? Caption needs to define this delta.

Figure 9b. What is Nb? Caption needs to define this.

P20 Line 19. "...the model better captured the seasonality of GPP and LE." This needs to clarify that this is a comparison between CAN and CA-RS, I presume. However, this statement does not necessarily appear to be true for CAN-RS versus TRUNK, which needs to be stated. But suppose that it is true that CAN-RS is a better predictor of seasonality than this parameterization of TRUNK, then could TRUNK be improved with better tuning? Does the TRUNK soil water stress function contain a parameter like m_psi in Eqn 3, and could this parameter be tuned to shift the strength of where the soil water stress occurs? If so, the implications of this hypothesis should be discussed

relative to the CAN-RS hypothesis and our understanding of the physiology.

P21 Line 34. Fig 18a. Should it be Fig17a?

P22. Line8. "functional diversity needs to be accounted for." This no longer needs to be a suggestion as Xu et al. (2016), Powell et al. (2018) and Anderegg et al. (2018) demonstrated this.

P22. Figure 17b. AGB versus precipitation has also been presented in Good et al. (2011) and Levine et al. (2016). The novelty of 17b is that the pattern is reproduced with a different hypothesis about how to represent demography and plant water stress compared to the other two studies. That is an important result and the differences between the hypotheses should be highlighted to inform our thinking about the ecology of this region.

Supplemental Table B1. What is PFT2?

Supplemental Table B1. K (recruitment parameter) is not in Eqn. 2 on page 4.

.

References:

Anderegg et al. (2018) Nature. doi: 10.1038/s41586-018-0539-7

Fisher et al. (2018) Global Change Biology. doi: 10.1111/gcb.13910

Good et al. (2011) Journal of Climate. doi: 10.1175/2010JCLI3865.1

Levine et al. (2016) PNAS. 10.1073/pnas.1511344112.

Miller et al. (2011) PNAS. doi: 10.1073/pnas.1105068108

Powell et al. (2018) New Phytologist. doi: 10.1111/nph.15271

Xu et al. (2016) New Phytologist. doi: 10.1111/nph.14009

Zhang et al. (2015) Global Change Biology. doi: 10.1111/gcb.12903

---

## Referee Comment (RC2) · Anonymous Referee #2 · 15 Jan 2019

The manuscript by Joetzjer et al presents a modeling study that makes a few relatively small changes to ORCHIDEE-CAN (CAN) and evaluates these changes against site-level and regional gridded data from the Amazon. The two (as far as I can tell) changes made to CAN are a recruitment function and a function that allows flexibility in root water uptake. Specifically, root-zone matric potential is calculated in CAN using a weighted mean where weights are based on root biomass in a soil layer, while the modifications (labeled CAN-RS) base the weights on the maximum amount of water that can be absorbed in a soil layer. Model evaluation of biomass accumulation following biomass removal shows that CAN and CAN-RS simulate improved above-ground biomass vs age since disturbance. Site-level daily and monthly fluxes (NEE, GPP, LE)

were simulated equally by the original model version (labeled TRUNK) and CAN-RS. CAN-RS primarily improves upon failures in CAN compared with TRUNK. TRUNK and CAN-RS held similar biases in simulating regional fluxes, albeit TRRUNk was lightly less biased at simulating ET while CAN-RS was perhapos slightly better at simulating spatial variability in mean annual GPP.

Overall this paper seems like a fairly small, incremental development to ORCHIDEE with little gain in model skill compared with TRUNK. The addition of the recruitment model and the change to root water uptake seem unrelated. While nothing is really incorrect methodologically, it's quite a long paper to wade through and I'm left wondering what has been learned. I disagree that the claim in the title: "importance of tree demography and root water uptake for modelling the carbon and water cycle of the Amazon" has been demonstrated with the current set of evaluations presented in the manuscript. Furthermore the paper lacks clear objectives and a number of the conclusions have not been demonstrated by the study. The study would benefit from a clear set of objectives (usually laid out in the final paragraph of the introduction). The study conclusions then need to be tied to these objectives. The conclusions on ln 21 pp 22 to ln 4 pp 23 are not based on the results of the study but are more "future directions" that come with very little background. These would be better placed in the discussion, with more surrounding discussion that gives a justification for the future work, or deleted.

To improve the chance of this study being cited I suggest focusing in on the few defined objectives, explain the model differences, and do a better job of demonstrating the improvements. The improvement in above-ground biomass simulation following disturbance is clear, though both CAN and CAN-RS are equivalent. But it is unclear to me what has caused this improvement. Is it that tree mortality is better simulated by CAN, or is it that carbon allocation to wood is better simulated? An analysis that can parse these two possible processes would be good. With regards to the improvement in the "seasonality of GPP and LE" (ln 18 pp 20), I'm not convinced. Do the authors mean CAN-RS compared with CAN or TRUNK? This improvement needs to be demonstrated

clearly.

Nutrients were alluded to in the discussion. ORCHIDEE has a version with nutrients enabled (Goll et al. 2017), why was that version not used for the current developments? Perhaps more to the point, the Amazon specific soil property maps developed by Marthews et al. (2014) were mentioned but were not used in this study. Why not? Tropical clay soils have very different hydraulic properties compared with temperate clay soils and the use of the USDA soil classification system seems like the wrong choice for this study.

Framing ORCHIDEE-CAN as a second generation DGVM is not entirely correct. All of the models described by Fisher et al. (2018) represent vertical competition for light allowing PFTs to compete. Each cohort has separate GPP and NPP. If I understand correctly, CAN and CAN-RS does not allow PFTs to compete, cohorts of a PFT share the same GPP, and parameterization is with map of PFT or species' distributions, which is not a property of a second generation DGVM.

In summary: Define a clear set of objectives A better link needs to be made between the root water uptake modification and the recruitment model modification/addition. The model evaluation ought to be in the context of the objectives with a clear explanation of why the various instances of the model differ (e.g. ABG∼year after disturbance, is it mortality or allocation?). A better case needs to be made for the improvements gained from the root water uptake modification. This is an interesting modification, I'm surprised that it doesn't perform better than TRUNK in most cases. Why not? Unless there is solid justification why Marthews soil properties were not used in this study, Marthews should be used. A more focused discussion and conclusion centered around the objectives is needed. I think the manuscript would benefit from an attempt at editing and consolidation to reduce the length and number of figures.

Minor comments:

ln 31 pp 1, the model reproducing fluxes as well as the original model is not really an

advance

ln 35-36, pp 1 this claim is not supported by the data in the paper

ln 2 pp 2, the amazon is already experiencing longer dry seasons in some places

ln 28 pp 2, suggest changing "density of root tissue" to "density of roots"

ln 5 pp 3, This study does not explore "relative contributions of . . . ," that would require a variance decomposition or similar methodologically

ln 30 pp 3, APAR does not decrease exponentially with LAI, rephrase

ln 4 pp 4, If I understand Naudts et al. (2015) the canopy structure is a statistical representation of a 3D canopy, not an explicit one. This distinction should be made clear

ln 10 pp 4, what does "this" refer to

ln 23 pp 4, Are these parameter values from Naudts et al. (2015), or were the changed based on Briennen for this study? I think more generally a table that makes clear the key differences between TRUNK, CAN, and CAN-RS would be useful. This could also include a brief description of processes important to interpreting the results, e.g. how LAI is calculated.

ln 24 pp 4, Could you add a brief description about how the number of individual (N) is simulated? Presumably if N exceeds Nmax then N is reduced to Nmax and this is self thinning. Are there other mortality processes? How does TRUNK simulate mortality?

ln 1-2 pp 5, It doesn't? How are LAI and canopy gaps related in CAN?

ln 20 pp 5, why don't you use the original notation from van Genuchten? It's helpful to use consistent notation.

ln 8 pp 6, while I understand the need to distinguish the different instances of the model I wouldn't describe the minor changes here as different versions of the model.

ln 25 pp 7, Where are these met data from? Can you cite a source or dataset?

Table 1 pp8, why was spin-up CO2 set to 370 ppm, this seems like a strange equilibrium condition.

Table 1 pp8, Am I right in thinking M34 is also described in the literature as K34? This site already has several names, I think it might be better to stick with K34.

ln 5 pp 11, suggest rephrasing "equal correlation extends ..."

Figure 3 pp 11, Why not also have Taylor plots of monthly data?

Figure 4 & 5 I don't find the diel cycles per month that useful or easy to read. I suggest focusing in on just a few key ones that demonstrate the model differences. Can uncertainty be added to the observations?

ln 3-4 pp 17, Can you be sure of this statement, how do you know?

Figure 10, Bias seems worst in CAN-RS

Figure 14, Why is TRUNK not shown?

ln 18-20 pp 19, How are kcmaint and CUE related, from the description it seems CUE = 1 – kcmaint.

ln 20 pp 22, TRUNK doesn't model diameter size distribution

Fisher, R. A., C. D. Koven, W. R. L. Anderegg, B. O. Christoffersen, M. C. Dietze, C. E. Farrior, J. A. Holm, G. C. Hurtt, R. G. Knox, P. J. Lawrence, J. W. Lichstein, M. Longo, A. M. Matheny, D. Medvigy, H. C. Muller-Landau, T. L. Powell, S. P. Serbin, H. Sato, J. K. Shuman, B. Smith, A. T. Trugman, T. Viskari, H. Verbeeck, E. Weng, C. Xu, X. Xu, T. Zhang, and P. R. Moorcroft. 2018. Vegetation demographics in Earth System Models: A review of progress and priorities. Global Change Biology 24:35–54. Goll, D. S., N. Vuichard, F. Maignan, A. Jornet-Puig, J. Sardans, A. Violette, S. Peng, Y. Sun, M. Kvakic, M. Guimberteau, B. Guenet, S. Zaehle, J. Penuelas, I. Janssens, and P. Ciais.
2017. A representation of the phosphorus cycle for ORCHIDEE (revisionÂă4520). Geosci. Model Dev. 10:3745–3770. Marthews, T. R., C. A. Quesada, D. R. Galbraith, Y. Malhi, C. E. Mullins, M. G. Hodnett, and I. Dharssi. 2014. High-resolution hydraulic parameter maps for surface soils in tropical South America. Geosci. Model Dev. 7:711–723. Naudts, K., J. Ryder, M. J. McGrath, J. Otto, Y. Chen, A. Valade, V. Bellasen, G. Berhongaray, G. Bönisch, M. Campioli, J. Ghattas, T. De Groote, V. Haverd, J. Kattge, N. MacBean, F. Maignan, P. Merilä, J. Penuelas, P. Peylin, B. Pinty, H. Pretzsch, E. D. Schulze, D. Solyga, N. Vuichard, Y. Yan, and S. Luyssaert. 2015. A vertically discretised canopy description for ORCHIDEE (SVN r2290) and the modifications to the energy, water and carbon fluxes. Geoscientific Model Development 8:2035–2065.

---

## Author Comment (AC1) · 13 Mar 2019

**Reply to Anonymous Referee #1 on "The importance of tree demography and root water uptake for modelling the carbon and water cycles of Amazonia".**

Dear Editor and Reviewer,

We thank you both for your efforts and time. We performed revisions on our paper according to your comments and we hope you find our changes and responses satisfactory. Please find below the reviewer comments (in bold font) and our replies (in normal font). In addition to the changes described below, we have revised the manuscript in several instances throughout the text and included a version of the manuscript with tracked changes. In the response to the reviewers, we pointed the lines in the revised version of the manuscript (without track changes).

As mentioned by both reviewers the paper is quite long and the objectives poorly defined, leading to some confusions.

We added in the introduction (p3 L11)
In this study, we test:

(1) the effect of representing layer-to-layer heterogeneity of soil water availability for trees and test if this process improves DGVMs carbon and water fluxes representation, by comparing CAN-RS and CAN

(2) the effect of representing demography and forest structure by distributing NPP into several classes of diameter, including self-thinning mortality and recruitment by comparing CAN and TRUNK

(3) the combined effect of including layer-to-layer heterogeneity of soil water and forest demography by comparing CAN-RS with TRUNK against observations."

The authors

**Anonymous reviewer #1**

**The manuscript by Joetzjer et al. explores how two new formulas for the ORCHIDEE dynamic global vegetation model influences carbon and water budget predictions of three Amazon forests. ORCHIDEE-DGVM, called TRUNK in this manuscript, is used as the baseline model and represents the land surface physiology as an aggregated 'big leaf'. The first new formula, ORCHIDEE-CAN (CAN), represents tree demography and**

aboveground mechanistic water transport through the stem, and is parameterized specifically for Amazon forests. The second new formula (CAN-RS) modifies CAN to include mechanistic soil water uptake by the roots. Improving the representation of belowground processes in land surface models is at the forefront of discussions in the modelling community and is a recommendation in the recent paper by Fisher et al. (2018), which is an authoritative review paper about the state-of-the-science of second generation dynamic vegetation models. I read this manuscript with great enthusiasm since these two formulas, CAN and CAN-RS, are at the cutting edge of model development needs. I do feel that this manuscript has great potential to add to our scientific knowledge about individual tree water-use strategies and physiology. However, in my opinion this manuscript is still too premature to be considered for publication. I support my opinion with the following

**General comments.**

1. My main critique of this manuscript, and one that I feel should be sufficiently addressed before it is considered for publication, is that it presents three alternative hypotheses about biological controls over tropical forest carbon and water cycles, yet these hypotheses are not clearly set up in the Introduction, described in the Methods, or convincingly interpreted in the Results and Discussion (comments on each section are detailed below). Also, given the results, the manuscript does not convincingly justify the asserted need to increase the TRUNK model complexity specifically with the new CAN-RS formulation.

**Introduction.**

(a) I recommend eliminating or reducing the first paragraph to a single sentence in order to create more space to describe the three hypotheses being evaluated.

(b) It is not clear what hypothesis TRUNK represents that is relevant to this study. I presume it has something to do with spatial aggregation; but since there is only a single line (P3, ln7) written about TRUNK, it is unclear what CAN and CAN-RS are being compared to. It would be helpful to have some context about where TRUNK has been a good hypothesis for explaining forest function, and carbon and water exchanges, and in what cases the hypothesis is not well supported.

(c) I feel that the Introduction will flow better if the three hypotheses that are all jointly introduced at the end of the Introduction are instead introduced in each's paragraph that contains its respective problem statement. For example, in paragraph three of the Introduction, (P2, ln23-29), it would be clearer if the proposed remedy (i.e. increased "layer-to-layer heterogeneity in soil-to-root resistance) is included as the concluding sentence of this problem-statement paragraph.

(d) It would be helpful to articulate in the Introduction what "fingerprints" the data are

**expected to have that would support of each particular hypothesis. E.g. what fingerprint (or marker) should be in the SWC data to support the given dynamic-root-water-uptake hypothesis? (Conversely, if dynamic root-water uptake is not important, what should the SWC data look like?) Articulating the fingerprints gives a justification for the selection of each benchmark presented. As the paper is currently written, the reader is first introduced to the specific benchmarks in the Results and has to intuit the rationale for each.**

We partly rewrote the introduction in line with your comments:

(a) We agree and shortened the first paragraph into one sentence. (p2 L3)
(b) We clarified the modeling hypothesis of the TRUNK version as well as our motivation for using it: p3 L19 "As a reference version, we used the TRUNK version updated for the CMIP6 exercise (Peylin et al., *in prep*) and widely used in global carbon cycle studies. TRUNK is based on Krinner et al. (2005) uses a "big-leaf" approximation, and simulates biomass dynamics from the allocation of NPP to leaves, wood and roots with a constant mortality. Woody biomass in the TRUNK is thus described by a single well mixed pool. Water uptake by roots in TRUNK is calculated by weighting a static root profile discretized upon 11 soil layers by soil moisture in each layer. CAN has the same photosynthesis model than TRUNK but includes a simplified forest demography model with 20 diameter classes upon which stand level GPP is distributed unevenly to favor high diameters, and mortality being the result of light competition (self-thinning). CAN-RS is equal to CAN but has a new root water uptake model described in this study". Also, we added in Section D (SI) the description of soil water stress representation in TRUNK.
(c) While we still introduced the three model versions in the same paragraph, we clarified the hypotheses tested as well as the version used (in order to test the hypothesis).

"We evaluated the effect of representing layer-to-layer heterogeneity of soil water availability for trees and test if this process improves DGVMs carbon and water fluxes representation." P2L26

"In this study, we propose to evaluate and test the benefit of an intermediate complexity second generation DGVM that represents demography and forest structure by downscaling NPP into several size classes and simulating mortality from tree density exceeding a threshold by killing preferentially a fraction of the smaller size classes, based on self-thinning principles."p2 L37

We evaluated (1) the effect of representing layer-to-layer heterogeneity of soil water availability for trees and test if this process improves DGVMs carbon and water fluxes representation by comparing CAN-RS and CAN (2) the effect of representing demography and forest structure by distributing NPP into several classes of

diameter, including self-thinning mortality and recruitment by comparing CAN and TRUNK and (3) the combined effect of including layer-to-layer heterogeneity of soil water and forest demography by comparing CAN-RS with TRUNK against observations. P3 L11

(d) We agree that we should introduce the benchmarks in the context of the hypothesis we are testing. We decided to add in the introduction "By comparing simulations by these three versions of the same DGVM over Amazonia against forest inventory data for biomass and stand characteristics, local site level flux tower measurements of carbon and latent heat exchange, and regional water and carbon fluxes and stocks observation-based models, our results shed light ...." P3 L16 and to be more specific in the method/results sections "At K67, a vertical soil moisture profile was available [*Nepstad et al.*, 2007] and used to test the soil water temporal dynamics in the CAN-RS version. P7 L24" and "We used tree diameter and height measurements (for 1592 trees) from the 2014 inventory on a 6.25 ha plot in order to evaluate CAN and CAN-RS forest structure representation".p7 L26

**Methods.**

**(a) Eqns. 3,4, 5 and 6 need to include an index (e.g. i) for each variable that is calculated separately for each soil layer (for example, if psi_s at P5 ln10 is calculated by layer, it should be psi_s,i, and i needs to be defined).**

Layer dependent variables or parameters are indicated, for example $\Psi_s(l)$ . We clarified the text by adding information: where $L$ is the number of layers (L=12) and $l$ the index of the layer considered. (p5 L7)

**Similarly, it is not clear how the model differentiates between the resource-use of different-sized trees. Should there be an index (e.g. j) in Eqn 5 for each size class of trees? Or, are all the roots uniformly distributed such that the seedlings have the same rooting depth and access to water as the canopy trees? Please clarify how this works. If it is the latter, then the new hydrodynamic physiology is not mechanistically linked to demography.**

The rooting depth is not explicitly associated to each cohort and the fine root biomass represents the sum of the fine root biomass of all cohorts. The coupling between above-ground demography and belowground water availability exists because for a given location, younger stands will have lower root biomass than older stands. It is, however, not fully mechanistic because root mass is calculated at the stand rather than at the cohort level. Considering that the smallest cohort reaches 6 m height (Fig.8) and the fact that we use a 4 m soil depth, we argue that this approximation is reasonable. Note that the aboveground hydraulic scheme, as described in Naudts et al. (2015) is cohort-dependent. However, your comment is very relevant and we clarified the modeling hypothesis in the method: "In CAN, $Biomass_{froots}$ represents the sum of fine root biomass of all

the cohorts calculated …" p5 L37).

**(b) I feel that the language related to the demography hypothesis needs to be much more precise. It is important to distinguish between whether the demographic rates emerge or are prescribed. If the demographic rates emerge due to a mechanism, then what is the hypothesized mechanism (C-starvation, hydraulic failure, etc.)? If the demographic rates are prescribed by the user, then this model does not simulate self- thinning through competition for limiting resources (P4 ln18) and it does not simulate demography as suggested in the Methods. Rather, the model represents the outcome of competition and self-thinning by prescribing a number-density by size-class distribution. In other words, does the model neglect all of the size-dependent internal dynamics of the ecosystem that give rise to the demographic rates from the bottom-up? If that is the hypothesis, then it needs to be clarified. Eqn 5 will act differently on individuals of different sizes due to non-linearities, than it will on all of the roots aggregated together. Making this distinction has very important implications for how ecologists and physiologist interpret the results of the simulations.**

In CAN, demography is not a fully emergent property (as in CLM-ED for example), but it is not fully prescribed either. Demographic growth rates for different cohorts emerge from downscaling stand-level NPP according to the rule from [*Deleuze et al.*, 2004], that partly represent size-dependent internal dynamics. Further, as explained in the manuscript (Fig.1 and Eq.1 and 2) recruitment of trees into the youngest cohort depends on the total amount of light reaching the forest floor and thus being transmitted through the canopy, given existing cohorts at each time step (Eq. 5). Mortality from light competition is based on an empirical PFT-specific self-thinning function that differentiates between cohorts, a parameterized by (Eq.4). Size-class (cohort) internal dynamics in CAN does not consider functional diversity and all size classes are defined with the same PFT specific parameters. We clarified both the hypothesis of the model and our motivation to test it in the introduction: "There are several approaches to represent tree demography in models. In some models forest structure and tree demography emerge from a mechanistic representation of competition and recruitment schemes with different species or groups of species having different functional traits. This method has given insights on tropical forests dynamics and resilience [*Zhang et al.*, 2015; *Levine et al.*, 2016; *Xu et al.*, 2016; *Fisher et al.*, 2018]. However they come with complex parameterizations, high computational cost and are not practical to run at global scale. In this study, we propose to evaluate and test the benefit of an intermediate complexity second generation DGVM that represents demography and forest structure by downscaling NPP into several size classes and simulating mortality from tree density exceeding a threshold by killing preferentially a fraction of the smaller size classes, based on self-thinning principles. Self-thinning is here parameterized but has been observed as an emerging property in tropical stands [*Pillet et al.*, 2017]." In addition we added the following sentences in the discussion, "CAN-RS has similar performance than the big-leaf / single biomass pool TRUNK version to represent forest fluxes, stocks and dynamics (Figs. S1,S3,S5,12,13), despite its higher complexity and more realistic description of soil water

uptake and demography. This is a positive result since a model like TRUNK is easier to adjust and intrinsically more stable, but less informative. For instance, adjusting the constant mortality parameter in TRUNK allows to match the observed biomass well, but lacks a mechanistic basis. CAN-RS offers the advantage to be assessable against other key measurable forest stand variables than total biomass, such as height and diameter classes for different forest ages (e.g. *Joetzjer et al.*, [2017]). Nevertheless, since demography parameters in CAN-RS are set constant for a single PFT describing all evergreen tropical forests, spatial variability of growth rates and mortality across the Amazon remains rather uniform compared to observations. Additional processes such as climate driven mortality and nutrient (phosphorus) limitation on growth leading to the prevalence of species with different functional traits across the Amazon would need to be included in the future development of this model. (L44 p17)

**(c) The soil-water stress function for TRUNK needs to also be included. Is it similar to Eqn 3?**

We don't think it is necessary to give the details of the TRUNK water stress function in the manuscript since we used the TRUNK as a reference to compare the carbon and water fluxes. To discuss the mechanism of soil water uptake we compare CAN-RS to CAN. It doesn't make sense to compare the mechanisms with TRUNK, because as now explained in the manuscript, TRUNK uses a simple multiplicative water stress factor on $V_{CMAX}$ and $g_s$ to represent water stress.

However, for clarity we added a section (section D) in the supplementary material, that is referred to in the text.

The stress function applied on the carboxylation capacity ($V_{CMAX}$) and on the stomatal conductance($g_S$) is derived from de Rosnay et al. (1999). The root density profile is defined as:

$$R(l) = e^{-cl}$$

This equation is discretized and we compute a normalized root profile for each soil layer $l$, $nroot(l)$.
No soil hydric stress is assumed when the soil moisture content in layer $l$ is larger than half the soil moisture difference between the wilting point $SWC_w(l)$ and field capacity $SWC_f(l)$. The limit value of soil moisture under no stress is defined by:

$$SWC_{nostress}(l) = SWC_w(l) + 0.5(SWC_f(l) - SWC_w(l))$$

The water stress index per layer $l$ is then calculated in each time step as:

$$u_s(l) = \frac{SWC(l) - SWC_w(l)}{SWC_{nostress}(l) - SWC_w(l)} nroot(l)$$

The total water stress in the soil column applied to $V_{CMAX}$ and $g_S$ is obtained by integration over the entire soil depth:

$$water\_lim = \sum_{l=1}^{L} u_s(l)$$

**(d) I recommend that all parameters that are new or have new values introduced in this study be placed in a Table that includes the appropriate reference. It is a little cumbersome to have to search the text for this information.**

We understand your point, and propose to add a table in the supplementary material as the paper is already long.

**Results and Discussion**

**(a) I am not convinced by the interpretation of the results for CAN and CAN-RS in terms of the claims that this paper is trying to make. The title implies that demography (CAN) and root water uptake (CAN-RS) are important for modelling tropical forest carbon and water cycles. The Abstract claims that modeling root water uptake in greater detail (CAN-RS) improves model performance (P1, ln32-35).**

We agree with your point and we changed the title and clarified the text in the revised version (the abstract and the concluding remark p18 are entirely rewritten). The simple description of forest demography in CAN does not improve stock and fluxes representation compared to the TRUNK, but it allows to model stand variables (diameter, height distributions) that are just ignored in TRUNK. We do not expect an automatic improvement of the performance of CAN-RS against TRUNK and we view as positive result that the more realistic CAN version has in fact similar performance than the easy-to-tune TRUNK version, while allowing to evaluate the CAN output against a new set of measurable variables that are known to be important to understand forest carbon dynamics. We refer to the citation by C. Prentice, which justifies the inclusion of known important processes and calls for observations to evaluate those new processes, which is the approach we have followed.

*"Although it seems reasonable to expect that a model including a larger subset of processes that are known to be important should be more realistic than a simpler model, increases in reliability and robustness are by no means automatic." (Prentice et al. 2015).*

Concerning the root water uptake scheme, CAN-RS and TRUNK show similar performances. However, TRUNK uses a simplistic formulation of soil water stress on stomatal conductance through Vcmax (see section D in SI), while in CAN water supply is calculated via a more realistic hydraulic architecture inspired from Hickler et al. (2006). Having a mechanistic representation of soil water stress is crucial for DGVMs to understand the vegetation response to drought. Our point here is to show that in normal conditions, CAN-RS performs as well as the

TRUNK, but because of its mechanistic approach CAN-RS has more potential, for instance to simulate hydraulic failure (and possibility mortality occurring from it). Comparing CAN to CAN-RS, showed that trees preferentially use water in the deepest soil layer during the dry season which led to improve LE and GPP seasonality for the soil types with a constraining soil water retention curve. Thus, if we acknowledge that CAN has a more realistic representation of tree growth than TRUNK and should be used in future studies, there is a clear improvement from the new root water uptake model of CAN-RS compared to CAN.

**The Results, Discussion and Conclusions make similar claims throughout. I do not feel that these conclusions are supported by the evidence presented in the Results for two primary reasons.**

**First, the Results need to give a much more thorough description of the Taylor Diagram (Figure 3); three sentences is not adequate (P10, ln8-12). Contrary to the Title and Abstract, the Taylor Diagram in Figure 3 indicates that TRUNK is an equivalent, but likely better, predictor of LE, GPP and NEE than CAN or CAN-RS. This is a highly significant outcome, yet it seems to be downplayed.**

We acknowledge in the paper that TRUNK performs well, but insist on the fact that TRUNK does not have a mechanistic representation of plant hydraulics. In the revised version we discussed the implication of these two different approaches.

**In all three panels, TRUNK has better or equivalent RMSE and correlation scores compared to the two other models. CAN and CAN-RS seem to only score better than TRUNK for the standard deviation metric. RMSE and correlation are indicators that the model is getting the pattern, and hence the mechanism, correct, whereas standard deviation indicates how well the mechanism is tuned to correctly capture the magnitude of the response. Therefore, for this type of analysis, which explores mechanistic controls, RMSE and correlation are more important indicators of model performance; yet the Results focus primarily on standard deviation. One objective way forward is to assign a skill score to each model as proposed by Taylor (2001) in Eqn 4 or 5. If a "penalty" is imposed, as in Eqn 5 of Taylor (2001), then the justification for the penalty needs to be described in the Methods.**

We modified this part of the manuscript. First, we want to compare CAN to CAN-RS to test the hypothesis of the root water uptake mechanism implemented on top of a realistic forest demography. " CAN-RS outperformed CAN at two of the sites for three metrics displayed in the Taylor diagram, K67 and GFG, but not at M34 where the two models had a similar performance (Fig. 3)." p11 L12

**Second, in just about all cases, TRUNK appears to be a better predictor of the bench-marks than either CAN or CAN-RS (Figures 3, 4, 5). To begin with, TRUNK seems to**

**match the observations in Figure 4 the best of the three models.**

Well, TRUNK and CAN-RS are very similar in terms of performances. CAN-RS outperforms TRUNK at GFG (fig. 4 and 5) but, as explained in the manuscript, this does not appear in the Taylor diagrams correlation (computed with hourly time-series) because of the mid-day depression simulated by CAN-RS. Mid-day depression in the model is related to the representation of tree hydraulics and should be recalibrated.

**CAN seems to be the weakest predictor, which tells me that either demography itself is not important, or the hypothesis contained in CAN about how to represent demography is not supported. Which is it and why?**

See our previous response to your point (a).

**Next, both CAN and CAN-RS show a midday suppression in GPP during the dry season that is not present in the observations (Fig. 5); however, TRUNK does not show the midday suppression. This tells me that the mechanistic water transport hypothesis as it is represented in CAN and CAN-RS is not supported by the observations.**

The midday depression in CAN and CAN-RS is related to the tree hydraulic scheme, and not to the new root water uptake (For example, they simulate a non-observed midday depression (Fig. 5), indicating that the above-ground hydraulic path needs to be revisited.). p17 L17 Again, thanks for this comment.

**In Figure 6, neither CAN nor CAN-RS reproduce observed SWC in any credible way. I am curious about what fingerprint the authors would expect the data to have in support of the CAN-RS hypothesis (this should be mentioned in the Introduction). Does Fig.6b actually possess such a fingerprint? On the other hand, the observations presented in Fig. 5 do not possess the expected fingerprint (i.e. midday suppression). If the observations do not possess the expected fingerprints, then what is the rationale for including the CAN-RS hypothesis in TRUNK in the first place? And, what is the rationale for using the eddy flux and SWC data as benchmarks if they do not contain the relevant fingerprints?**

In addition to test the "demographic" and "soil water uptake" hypothesis, the paper also aims at evaluating CAN-RS ability to reproduce (at least as well as the reference version TRUNK) the main expected features of a DGVM, that include its ability to correctly simulate carbon and water fluxes and stocks. With the flux-tower data we are able to show that CAN-RS outperforms CAN in term of RMSE and correlation, at certain soil types. We discussed the midday depression simulated by CAN (and CAN-RS). Looking at soil water content measurements allow us to evaluate the seasonal dynamic pattern (wet/dry season and root water uptake). At K67 during the 2003 dry season, there is a mismatch between simulated (CAN-RS) and observed fluxes. The model has insufficient recharge of the soil during the wet season, leading to a dry bias compared

to observations. This suggests that the mismatch in fluxes is not related to the water uptake scheme, but rather to the soil water vertical transport, or to missing processes such as groundwater recharge. We clarified the manuscript: "Since the 2003 wet season was drier (1276 mm) than the ones in 2002 (1683 mm) and 2004 (1849 mm) (Fig. 6a), the amount of precipitation was insufficient to recharge the soil after dry-season depletion in the model (Fig. 6c-d) but not in the observations (Fig. 6b) explaining the mismatch between the observed and simulated fluxes (Fig 4 and 5). This too slow recharge of the soil in the wet season may be due to underestimated vertical infiltration (e.g. ignored soil conductivity increased by roots in weathered tropical soils) or to the lack of groundwater storage mechanisms" p11 L23

**I do wonder if the observed SWC data presented in Fig. 6 has the correct resolution to be a valid test for the CAN-RS hypothesis. If not, then the authors might consider Miller et al. (2011), it contains the fingerprint that supports the CAN-RS hypothesis (Fig. 1b) and perhaps would be a better test of the CAN-RS hypothesis.**

Thanks for the reference. However, as discussed previously we were interested in the temporal dynamics of the soil water content to understand the mismatch between the observed and simulated fluxes.

**Finally, given that TRUNK has the correct pattern (and no midday suppression) but the incorrect magnitude in Figure 5, could it be that that the structure of TRUNK is correct, but it just needs better tuning?**

TRUNK is used as a reference, and does not include a realistic below and above ground water transport transfer. TRUNK representation of water stress on the stomatal conductance and Vcmax is a shortcut that can be tuned. The point was to show that adding a mechanistic below ground root water uptake model combined with tree hydraulics better matches observations (CAN-RS) than just tree hydraulics (CAN), pointing out the argument that trees can line water in the deepest layers when the soil gets drier. But, overall we agree, a simplistic model can give correct answer but for the wrong reasons.

**(b) The Taylor Diagram is certainly a good method for cross-model comparisons to observations, but one of its limitations that should be addressed in the Discussion is that, unlike AIC for statistical models, it does not account for the trade-off between model simplicity and complexity. As acknowledged in the Introduction (P3, ln3), process- based land surface models have extremely high degrees of freedom that can lead to compensating errors and equifinality. It is unclear if and how this trade-off has been considered in the conclusion that CAN-RS is an improvement over TRUNK.**

As we compare CAN-RS to CAN that broadly have the same structure and parameters, we don't think it is necessary. Again, TRUNK is only used as a reference.

**2. This manuscript would be much more impactful with greater synthesis with the many**

recent and important advances in the field of dynamic vegetation model development. The manuscript also needs greater synthesis about how these different hypotheses inform our physiological and ecological understanding of tropical forests.

(a) For example, Xu et al. (2016) and Powell et al (2018) also use a dynamic vegetation model that explicitly represents demography and mechanistic water movement through the soil-plant-atmosphere continuum. Xu. et al (2016) explores mechanisms related to water-stress avoidance, while Powell et al. (2018) explores mechanisms related to tolerance of water-stress. There is a good opportunity to connect the insights into belowground mechanisms explored in this study to the insights into aboveground mechanisms explored in those two studies.

(b) This study also addresses some of the issues raised in the recently published Anderegg et al. (2018).

(c) Figure 15. Figures of the spatial distribution of AGB across the Amazon basin have been widely published. Zhang et al. (2015) reported such figures for several DGVMs (Fig. 3s) including one like CAN that contains a size-structure and demography hypothesis. I feel that Figure 15 is a significant result because the CAN hypothesis does not show a strong spatial gradient in AGB, which is in contrast to two hypotheses in Zhang et al. (2015) that do capture the gradient reasonably well (see Figure S3f in Zhang et al.). Why is the CAN hypothesis not supported by the data, but the other two hypotheses are supported by the data? Is it due to CAN model structure, experimental design, parameterizations? What can we learn about the ecology and model development needs from the contrasting results?

(d) Finally, this study and Levine et al (2016) proposed two different hypotheses about how to represent demographic processes in a land surface models; and, the two stud- ies produces very important and contrasting results. Levine et al. (2016) argues that the size-structured model hypothesis is supported by the data because the demographic rates emerge from a bottom-up formulation of spatial heterogeneity. The CAN formulation, in contrast, appears to be a top-down scaling hypothesis. Given that the bottom-up approach agrees with benchmark tests (e.g. Xu et al. 2016, Zhang et al. 2015, Powell et al. 2018), but the top-down approach (CAN) does not perform better than TRUNK (a spatially aggregated hypothesis), what does this tell us about these approaches for representing demographic processes?

Making these connections will provide a much more complete picture about the con- tribution this study makes to the state-of-the-science regarding demographic dynamic vegetation models, modeling plant hydrodynamics, and understanding tropical forest ecology in general.

Concerning your points (a) and (b) it is somehow complicated to give insight from CAN-RS

compared to results obtained with ED2-hydro and ED. This two studies mainly explored above-ground hydraulic processes with a more sophisticated model than CAN-RS. Taking into account functional diversity for hydraulic traits to get realistic response to drought is feasible with a DGVM that considers co-existing groups of species and explicit competition. We agree that models should tend towards such a representation, but we feel it is beyond the objective of this study which aims at simulating long time scales and gridded fluxes and pools. Nevertheless, we included the references in the introduction. Concerning point (c) and (d) we clarified the manuscript about the modeling hypotheses in CAN. The goal of this study is to improve a global DGVM part of a land surface model for tropical forests in the Amazon by including a simple set of demographic equations and size-structured cohorts, based on generic equations previously applied to temperate forest biomes (Naudts et al., 2015). Detailed stand level models can be developed and benchmarked but usually, these models do not close the energy budget and do not resolve diurnal time scales, thus they cannot be coupled with Earth System Models. Here we show that the DGVM can account for size and height distribution together with biomass-age relationships in Amazon forests, which is not the case of the bottom-up models cited above.

**3. The total number of Figures could be reduced. The manuscript loses focus between Figures 10 and 17 and the rationale for these figures is not very well established in the Introduction (i.e. I do not see anything regarding predictions of spatial variability in the Introduction). I recommend this paper focus on figures that directly test the specific demography and root water uptake hypotheses contained in TRUNK, CAN, and CAN-RS. Figures 10 to 17 are summaries and their relevance is predicated on the CAN and CAN-RS hypotheses being more strongly supported by the data than the TRUNK hypothesis. Also, the Discussion needs to be better integrated with the figures presented in the Results. All key results figures should be cited in the Discussion; if they are not, then this tells me that they are not central to the story of the manuscript and should be moved to the Supporting Material.**

ORCHIDEE is a DGVM that is included in the IPSL Earth System Models model. It is designed to run globally and in a coupled mode with the atmosphere. One objective of this study is to demonstrate that CAN-RS performs at least as well as the TRUNK, locally and regionally, meanwhile being able to simulate realistic forest structure, which the TRUNK cannot do. Therefore, we think that keeping the regional evaluation figures in the manuscript is important. To make this point clearer we added in the introduction "By comparing simulations by these three versions of the same DGVM over Amazonia, against forest inventory and local and regional water and carbon fluxes and stocks observations, …" and "CAN-RS has similar performance than the big-leaf / single biomass pool TRUNK version to represent forest fluxes, stocks and dynamics (Fig 10-13,15,16 17), despite its higher complexity and more realistic description of soil water uptake and demography. This is a positive result since a model like TRUNK is easier to adjust and intrinsically more stable, but less informative. For instance, adjusting the constant mortality parameter in TRUNK allows to match the observed biomass well, but lacks a

mechanistic basis. CAN-RS offers the advantage to be assessable against other key measurable forest stand variables than total biomass, such as height and diameter classes for different forest ages" p17 L44 in the discussion. We agree that the paper is too long and decided to move Fig. 10, 11, 14, 15 to the supplementary material.

**Specific comments.**

**P1. Line 1. Title. The title highlights the importance of tree demography, yet the experiment is not set up to explicitly isolate this, as CAN, contains at least three differences from TRUNK: (1) explicit tree demography, (2) mechanistic stem water transport, (3) Amazonian specific parameterization. With the CAN versus TRUNK comparison, the authors need to evaluate how each of these differences (as well as any others that may be unmentioned) individually impact model predictions. Otherwise, attribution to any particular modification is confounded by its interaction with the other two modifications.**

We changed the title for « Effect of tree demography and flexible root water uptake for modeling the carbon and water cycles of Amazonia". See previous comments for the hypothesis tested and the versions used.

**P1. Line 22. "remains challenging". This is very vague and will mean many different things to many different people, much of which is unrelated to the subject of this manuscript. This needs to be clarified to keep the reader focused on the subject of this paper.**

**# Comments related to the abstract:**

**P1. Line 23. "These" lacks an antecedent and therefore it is unclear what the limitations are.**

**P1. Lines 25-28. Why evaluate these three hypotheses? The Abstract lacks a problem statement with sufficient detail to motivate their evaluation.**

**P1. Line 31. "as well as TRUNK. . .at local and regional scales. . ." If this is the case, then why use a more complex model? This statement also does not support the conclusion in line 34, ". . .improves the representation of biogeochemical cycles. . ."**

Improving the below-ground mechanism (CAN vs. CAN-RS) improves the representation of the biogeochemical fluxes. The revised version of the abstract should be clearer.

**P1. Line 35. Last sentence. This sentence is of course true. . .about a lot of things. However, this sentence is quite vague and therefore loses any specific meaning with respect to the key findings of this study. It should be revised to specify "the variation [in what aspect of] ecological functioning". Plant hydraulics? Soil hydraulics? Life history strategies? Phenology? Biogeochemical cycling? Life forms palms, shrubs, trees, lianas? Disturbance**

**regimes? Etc.?**

Indeed, thanks. We removed this sentence and insist on the ORCHIDEE-CAN limitation.

**Revised abstract:**

Global Dynamic global vegetation models (DGVMs) part of the land surface schemes of Earth System Models do not represent forest structure and demography. Further, those models usually treat water uptake by plants from soil bucket schemes or multiple layers soil models, with non-physical water stress functions. This situation is a source of uncertainty for modelling the current state and future dynamics of Amazonian forest. We included recruitment processes and added a physically based model of soil water limitation on root water uptake for tropical evergreen forests in ORCHIDEE-CAN, a DGVM with demography equations previously established for temperate and boreal forests. The model was re-calibrated against tropical forest inventory measurements in Guyana and Brazil, including biomass-age relationships, diameter and height distributions. We compared the output of two model versions, one with forest demography only (CAN) and the other with demography and hydraulic transport of water from soil pores to roots coupled to xylem conduction by stems (CAN-RS). The results of the CAN and CAN-RS versions are compared with those of the standard TRUNK version of the DGVM used for global applications, which follows a big-leaf approximation, does not resolve forest demography and has a constant mortality applied to a well-mixed biomass pool. Site-level observations of turbulent energy and $CO_2$ fluxes at flux tower locations are used to evaluate the three versions of the model, as well as measurements of carbon stocks and stand density at inventory plots. Gridded observation-based models output of photosynthesis (GPP) and evapotranspiration (LE) are used across the Amazon basin. CAN and CAN-RS can reproduce observed forest structure variables, including tree density and height / diameters distribution, which were not modeled by TRUNK. In CAN-RS, water uptake from tree roots sustained from the deepest soil layers during the dry season, significantly improved the modelling of seasonal photosynthesis and evapotranspiration variations compared to CAN, especially over the Guyana and Brazilian shields with prevailing clay soils. The performances of CAN-RS are shown to be equivalent to those of TRUNK for carbon and water fluxes and carbon stocks across Amazonia, despite new processes being added. This result indicates that forest demography and mechanistic root water uptake did not degrade the DGVM performance while offering a more realistic simulation of key processes for the Amazon forest.

**P2. Line 1. "will likely". This is too strong of a statement. Perhaps substitute " are predicted to".**

This paragraph has been removed from the manuscript.

**P2. Line 8. The "large-scale dieback" is a quite dated result that has been updated widely in the literature over the subsequent decade and a half. Consider revising to be more current with our understanding of the system.**

Same paragraph which has been removed from the manuscript.

**P2. Line 20. Xu et al. (2016) should be referenced here.**

Indeed, thanks.

**P5. Eqn 4. Is there a typo? Change min to max?**

Indeed, thanks for noticing it. We changed the equation

**P10. Table 2. AGB. I think it is worth noting that none of the models capture the observed spatial variability of AGB.**

The table 2 only summarized models' performances at the 3 sites.

**P15. Line 16. Exchange "slightly" with "by 33%" (the two are probably not equivalent). Figure 9a. y-axis. Delta of what? Caption needs to define this delta. Figure 9b. What is Nb? Caption needs to define this.**

We removed the "slightly" and modified the caption of the figure 9.

**P20 Line 19. ". . .the model better captured the seasonality of GPP and LE." This needs to clarify that this is a comparison between CAN and CA-RS, I presume. However, this statement does not necessarily appear to be true for CAN-RS versus TRUNK, which needs to be stated. But suppose that it is true that CAN-RS is a better predictor of seasonality than this parameterization of TRUNK, then could TRUNK be improved with better tuning? Does the TRUNK soil water stress function contain a parameter like m_psi in Eqn 3, and could this parameter be tuned to shift the strength of where the soil water stress occurs? If so, the implications of this hypothesis should be discussed relative to the CAN-RS hypothesis and our understanding of the physiology.**

We clarified that the improvement concerns the comparison between CAN-RS to CAN in the manuscript.

**P21 Line 34. Fig 18a. Should it be Fig17a?**

Indeed than for noticing it.

**P22. Line8. "functional diversity needs to be accounted for." This no longer needs to be a suggestion as Xu et al. (2016), Powell et al. (2018) and Anderegg et al. (2018) demonstrated this.**

We rewrote the concludings remarks (see p18)

**P22. Figure 17b. AGB versus precipitation has also been presented in Good et al. (2011) and Levine et al. (2016). The novelty of 17b is that the pattern is reproduced with a different**

**hypothesis about how to represent demography and plant water stress compared to the other two studies. That is an important result and the differences between the hypotheses should be highlighted to inform our thinking about the ecology of this region.**

Thanks for the references. However Good et al. 2011 and Levine et al.2016 look at the biomass sensibility to the dry season length (DSL). In Fig 17, we look at the average biomass gradient vs. annual precipitation. We agree that an analysis of biomass sensibility to DSL should be conducted in ORCHIDEE-CAN, but it is beyond the scope of this present study.

**Supplemental**

**Table B1. What is PFT2?**

It is evergreen tropical forest, we clarified the legend.

**Supplemental Table B1.**

**K (recruitment parameter) is not in Eqn. 2 on page 4.**

Indeed, we removed this line (as we put the number directly in the equation)

**References:**

**Anderegg et al. (2018) Nature. doi: 10.1038/s41586-018-0539-7**

**Fisher et al. (2018) Global Change Biology. doi: 10.1111/gcb.13910**

**Good et al. (2011) Journal of Climate. doi: 10.1175/2010JCLI3865.1**

**Levine et al. (2016) PNAS. 10.1073/pnas.1511344112.**

**Miller et al. (2011) PNAS. doi: 10.1073/pnas.1105068108**

**Powell et al. (2018) New Phytologist. doi: 10.1111/nph.15271**

**Xu et al. (2016) New Phytologist. doi: 10.1111/nph.14009**

**Zhang et al. (2015) Global Change Biology. doi: 10.1111/gcb.12903**

**Anonymous Referee #2**

**The manuscript by Joetzjer et al presents a modeling study that makes a few relatively small changes to ORCHIDEE-CAN (CAN) and evaluates these changes against site-level and regional gridded data from the Amazon. The two (as far as I can tell) changes made to CAN are a recruitment function and a function that allows flexibility in root water uptake. Specifically, root-zone matric potential is calculated in CAN using a weighted mean where weights are based on root biomass in a soil layer, while the modifications (labeled CAN-RS) base the weights on the maximum amount of water that can be absorbed in a soil layer. Model evaluation of biomass accumulation following biomass removal shows that CAN and CAN-RS simulate improved above-ground biomass vs age since disturbance. Site-level daily and monthly fluxes (NEE, GPP, LE) were simulated equally by the original model version (labeled TRUNK) and CAN-RS. CAN-RS primarily improves upon failures in CAN compared with TRUNK. TRUNK and CAN-RS held similar biases in simulating regional fluxes, albeit TRRUNk was lightly less biased at simulating ET while CAN-RS was perhapos slightly better at simulating spatial variability in mean annual GPP.**

**Overall this paper seems like a fairly small, incremental development to ORCHIDEE with little gain in model skill compared with TRUNK. The addition of the recruitment model and the change to root water uptake seem unrelated.**

In addition of the recruitment function and the new root water uptake scheme, we recalibrated the demography equations of CAN, previously established for mono-age temperate in Europe only. The modeled self-thinning mortality function, although well-verified for mono-aged temperate stands has been measured for tropical stands (Pillet et al., 2017) and was here applied for the first time to model mortality from light competition in tropical forests, with added recruitment. Recruitment is a fundamental process that maintains competition in tropical forests, resulting in high tree densities. This first version of CAN adapted to tropical forests was also evaluated for height and density distributions at Paracou (French Guiana) and for biomass stocks across the Amazon basin.

**While nothing is really incorrect methodologically, it's quite a long paper to wade through and I'm left wondering what has been learned. I disagree that the claim in the title: "importance of tree demography and root water uptake for modelling the carbon and water cycle of the Amazon" has been demonstrated with the current set of evaluations presented in the manuscript.**

We changed the title: "Effect of tree demography and flexible root water uptake for modeling the carbon and water cycle of Amazonia?" We also shortened the paper.

**Furthermore the paper lacks clear objectives and a number of the conclusions have not been demonstrated by the study.**

Thanks for this comment, which is also pointed out by reviewer #1. We agree and clarified the objectives for the revised version of the paper in the introduction.

> "We evaluated the effect of representing layer-to-layer heterogeneity of soil water availability for trees and test if this process improves DGVMs carbon and water fluxes representation." P2L26
>
> "In this study, we propose to evaluate and test the benefit of an intermediate complexity second generation DGVM that represents demography and forest structure by downscaling NPP into several size classes and simulating mortality from tree density exceeding a threshold by killing preferentially a fraction of the smaller size classes, based on self-thinning principles."p2 L37
>
> We evaluated (1) the effect of representing layer-to-layer heterogeneity of soil water availability for trees and test if this process improves DGVMs carbon and water fluxes representation by comparing CAN-RS and CAN (2) the effect of representing demography and forest structure by distributing NPP into several classes of diameter, including self-thinning mortality and recruitment by comparing CAN and TRUNK and (3) the combined effect of including layer-to-layer heterogeneity of soil water and forest demography by comparing CAN-RS with TRUNK against observations. P3 L11

We also rewrote the conclusions (concludings remarks p18, see below):

**The study would benefit from a clear set of objectives (usually laid out in the final paragraph of the introduction). The study conclusions then need to be tied to these objectives. The conclusions on ln 21 pp 22 to ln 4 pp 23 are not based on the results of the study but are more "future directions" that come with very little background. These would be better placed in the discussion, with more surrounding discussion that gives a justification for the future work, or deleted.**

We agree and changed the manuscript accordingly.

The simple description of forest demography in CAN does not improve stock and fluxes representation compared to the TRUNK, but it allows to model stand variables (diameter, height distributions) that are just ignored in TRUNK. We do not expect an automatic improvement of the performance of CAN-RS against TRUNK and we view as positive result that the more realistic CAN version has in fact similar performance than the easy-to-tune TRUNK version, while allowing to evaluate the CAN output against a new set of measurable

variables that are known to be important to understand forest carbon dynamics. We refer to the citation by C. Prentice (2015), which justifies the inclusion of known important processes and calls for observations to evaluate those new processes *"Although it seems reasonable to expect that a model including a larger subset of processes that are known to be important should be more realistic than a simpler model, increases in reliability and robustness are by no means automatic".* Concerning the root water uptake scheme, CAN-RS and TRUNK show similar performances. However, TRUNK uses a simplistic formulation of soil water stress on stomatal conductance through Vcmax (see section D in SI), while in CAN water supply is calculated via a more realistic hydraulic architecture inspired from Hickler et al. (2006). Having a mechanistic representation of soil water stress is crucial for DGVMs to understand the vegetation response to drought. Our point here is to show that in normal conditions, CAN-RS performs as well as the TRUNK, but because of its mechanistic approach CAN-RS has more potential, for instance to simulate hydraulic failure (and possibility mortality occurring from it). Comparing CAN to CAN-RS, showed that trees preferentially use water in the deepest soil layer during the dry season which led to improve LE and GPP seasonality for the soil types with a constraining soil water retention curve. Thus, because of a more realistic representation of forest structure and root water uptake than TRUNK, CAN-RS should be used in future studies.

**To improve the chance of this study being cited I suggest focusing in on the few defined objectives, explain the model differences, and do a better job of demonstrating the improvements.**

**The improvement in above-ground biomass simulation following disturbance is clear, though both CAN and CAN-RS are equivalent. But it is unclear to me what has caused this improvement. Is it that tree mortality is better simulated by CAN, or is it that carbon allocation to wood is better simulated? An analysis that can parse these two possible processes would be good.**

As explained in the manuscript AGB in TRUNK (p3 L5), like in any DGVM, biomass is described as a single well-mixed pool, can easily be tuned by changing a single mortality rate parameter. Oppositely, in CAN the stand level mortality rate is an emerging result of modelled growth and cohort-dependent turn over from self-thinning (Fig. 1). Thus, it is possible to match any AGB value in TRUNK by adjusting the average mortality rate but this does not represent a realistic process to kill trees.

**With regards to the improvement in the "seasonality of GPP and LE" (ln 18 pp 20), I'm not convinced. Do the authors mean CAN-RS compared with CAN or TRUNK? This improvement needs to be demonstrated clearly.**

Concerning the root water uptake scheme, CAN-RS and TRUNK show similar performances. However, TRUNK applies a non-physical water stress on stomatal conductance and photosynthetic rates (see section D in SI) irrespective of root biomass and tree conductivity. In CAN the water transport in stems (xylem) uses the hydraulic model described in Hickler et al. (2006) but water transport from soil pores to root is un-limited. In CAN-RS, we added a resistance for soil-to-root transport. Having a mechanistic representation of water stress in the continuum between soil, plant and atmosphere is crucial for DGVMs to account for vegetation response to drought. Our point here is to show that in non-stressed conditions, CAN-RS performs as well as the TRUNK for GPP and LE, but because of its mechanistic approach CAN-RS has more potential. Comparing CAN to CAN-RS showed that by taking in account soil-to-root transport processes, trees preferentially use water in the deepest soil layer during the dry season [*Moreira et al.,* 2000] which led to improve LE and GPP seasonality for the soil types with a constraining soil water retention curve. This result is interesting because it shows that to obtain a good match with observed GPP and LE fluxes, one needs to account for both stem and soil resistance in a model with hydraulics. We clarified the revised version of the manuscript in the introduction

**Nutrients were alluded to in the discussion. ORCHIDEE has a version with nutrients enabled (Goll et al. 2017), why was that version not used for the current developments?**

The version that includes nutrients does not include tree demography and hydraulic architecture. It is our objective to eventually merge the 2 versions.

**Perhaps more to the point, the Amazon specific soil property maps developed by Marthews et al. (2014) were mentioned but were not used in this study. Why not? Tropical clay soils have very different hydraulic properties compared with temperate clay soils and the use of the USDA soil classification system seems like the wrong choice for this study.**

We agree that, from a scientific point of view, a detailed soil maps (such as Marthews') should be used. However, the ultimate goal of ORCHIDEE-CAN is to run at global scale, and, Marthew's map coverage is limited to the Amazon.

**Framing ORCHIDEE-CAN as a second generation DGVM is not entirely correct. All of the models described by Fisher et al. (2018) represent vertical competition for light allowing PFTs to compete. Each cohort has separate GPP and NPP. If I understand correctly, CAN and CAN-RS does not allow PFTs to compete, cohorts of a PFT share the same GPP, and parameterization is with map of PFT or species' distributions, which is not a property of a second generation DGVM.**

We agree that the strategy for modeling demography is crucial. CAN has an intermediate complexity top-down parameterization of competition for light that accounts for size-cohorts with size-specific growth rates (from differentiated allocation of stand NPP) and vertical structure of

LAI given by the height of the different cohorts, but, indeed, CAN does not account for differentiated PFTs and plant functional traits among cohorts, not for the shifts in PFTs as observed in tropical forest. Note that cohorts share the same GPP but each cohort is allocated a different fraction of assimilates.

We added in the introduction: "There are several approaches to represent tree demography in models. In some models forest structure and tree demography emerge from a mechanistic representation of competition and recruitment schemes with different species or groups of species having different functional traits. This method has given insights on tropical forests dynamics and resilience [*Zhang et al.*, 2015; *Levine et al.*, 2016; *Xu et al.*, 2016; *Fisher et al.*, 2018]. However they come with complex parameterizations, high computational cost and are not practical to run at global scale. In this study, we propose to evaluate and test the benefit of an intermediate complexity second generation DGVM that represents demography and forest structure by downscaling NPP into several size classes and simulating mortality from tree density exceeding a threshold by killing preferentially a fraction of the smaller size classes, based on self-thinning principles. Self-thinning is here parameterized but has been observed as an emerging property in tropical stands [*Pillet et al.*, 2017]." P2 L31

**In summary: Define a clear set of objectives A better link needs to be made between the root water uptake modification and the recruitment model modification/addition. The model evaluation ought to be in the context of the objectives with a clear explanation of why the various instances of the model differ (e.g. ABG~year after disturbance, is it mortality or allocation?). A better case needs to be made for the improvements gained from the root water uptake modification. This is an interesting modification, I'm surprised that it doesn't perform better than TRUNK in most cases. Why not?**

CAN does not outperformed the TRUNK that has been tuned, optimized, fine tuned, and tested over and over during the past 10 years. The TRUNK may do sometimes better but for the wrong reasons (continuous mortality – no recruitment, regarding AGB processes, tuned water stress function regarding water stress). We rewrote the manuscript accordingly (see previous comments).

**A more focused discussion and conclusion centered around the objectives is needed. I think the manuscript would benefit from an attempt at editing and consolidation to reduce the length and number of figures.**

Thanks to both reviewers' comment, the revised version is more straightforward, shorter and we moved Fig. 10, 11, 14, 15 to the SI.

**Minor comments:**

**ln 31 pp 1, the model reproducing fluxes as well as the original model is not really an advance**

It is because, as explained, the TRUNK represents water stress using a simple function that has been tuned to reproduced the observation, while in CAN-RS, the water stress is represented by a mechanistic hydraulic architecture and root water uptake. See previous comments (concluding remarks).

**ln 35-36, pp 1 this claim is not supported by the data in the paper**

Indeed, we removed this sentence. See revised abstract in the reply to reviewer1

**ln 2 pp 2, the amazon is already experiencing longer dry seasons in some places**

Indeed. To shorter the manuscript and as suggested by reviewer 1, the introduction was partly re-written and this paragraph is not in the manuscript anymore.

**ln 28 pp 2, suggest changing "density of root tissue" to "density of roots"**

Thanks, we changed the text accordingly. (l24 p2)

**ln 5 pp 3, This study does not explore "relative contributions of . . . ," that would require a variance decomposition or similar methodologically**

We agree that this sentence was not appropriate and modified it for "This study explores the  importance of tree demographic and below-ground hydraulic processes on the Amazonian carbon and water cycles, by testing the… "

**ln 30 pp 3, APAR does not decrease exponentially with LAI, rephrase**

"Carbon assimilation is based on the leaf-scale equation of *Farquhar et al.*, [1980] for C3 plants and is assumed to scale from leaf to canopy with APAR (absorbed photosynthetically active radiation) decreasing exponentially with leaf area index (LAI), according to the big-leaf approximation"

**ln 4 pp 4, If I understand Naudts et al. (2015) the canopy structure is a statistical representation of a 3D canopy, not an explicit one. This distinction should be made clear**

The canopy structure is mechanistically simulated using downscaling of NPP from Deleuze et al., 2004 with calibrated parameters for the tropic and self-thinning and recruitments rules. The deeper description of the modeling hypothesis in CAN (see previous comment) should be clear in the revised version of the manuscript.

**ln 10 pp 4, what does "this" refer to**

It refers to transpiration; we changed it in the revised version to be clearer. (L42 p3)

**ln 23 pp 4, Are these parameter values from Naudts et al. (2015), or were the changed based on Briennen for this study? I think more generally a table that makes clear the key differences between TRUNK, CAN, and CAN-RS would be useful. This could also include a brief description of processes important to interpreting the results, e.g. how LAI is calculated.**

Adapting CAN to the tropics required changing several parameters, including self-thinning one's (indeed based on Briennen et al., 2015). We understand your point about making a table describing the 3 models. Now that we clarified the hypothesis behind each model's version, we don't think a table is necessary. However, we added a table (TableS2) in the supplementary that summarized all the parameters calibrated for CAN and CAN-RS.

**ln 24 pp 4, Could you add a brief description about how the number of individual (N) is simulated? Presumably if N exceeds Nmax then N is reduced to Nmax and this is self thinning. Are there other mortality processes? How does TRUNK simulate mortality?**

The number of individual at the beginning of the simulation is set at Nmax,ini (a parameter, see S2). Nmax,ini can be considered as the number of seeds. The number of individual depends on the Number of tree at the previous time step, and Nrecruits minus Ndead. In CAN the main processes that is actually killing trees is the self-thinning (in the Amazon). However, carbon starvation is also implemented in the model (and happens only under drastic conditions). In CAN, the number of individuals (N) decreased by self-thinning (Eq. 5) or increased by recruitment (Eq. 4) depending on the soil and atmospheric conditions.

As explained in the manuscript p16 l4 : "Contrary to CAN in which mortality is an emerging result of modelled competition processes via self-thinning (Fig. 1), in TRUNK the background tree mortality is very simply set as a constant fraction of the woody carbon pool, defined by a "residence time" parameter which is poorly constrained by observations [*Sitch et al.*, 2003]."

**ln 1-2 pp 5, It doesn't? How are LAI and canopy gaps related in CAN?**

Thanks for this comments. Indeed, a canopy gaps fraction is calculated in CAN based on Harverds et al., 2012. The gap fraction is after used as a control on the leaf carbon allocation coefficient (see eq. 6 in Naudts et al., 2015), therefore the fraction of gap indirectly influence LAI. However, the role of the gap fraction was negligible in our analysis. We clarified the manuscript by removing this sentence.

**ln 20 pp 5, why don't you use the original notation from van Genuchten? It's helpful to use consistent notation.**

Indeed, this is not the original notation of van Genuchten, but we decided to be consistent with ORCHIDEE's code for future users

**ln 8 pp 6, while I understand the need to distinguish the different instances of the model I wouldn't describe the minor changes here as different versions of the model.**

It is the same model (CAN) (as we state in the introduction), where we add a soil-to-root resistance, that we called CAN-RS.

**ln 25 pp 7, Where are these met data from? Can you cite a source or dataset?**

Thanks, the data from fluxnet can be download at http://fluxnet.fluxdata.org/, expect at Paracou, where a request to Damien Bonal is needed. We will provide a data availability paragraph for publication (requested by biogeoscience), as well as one for the code availability.

**Table 1 pp8, why was spin-up CO2 set to 370 ppm, this seems like a strange equilibrium condition.**

Thanks, indeed it's a mistake, spin-up was realized at 350ppm.

**Table 1 pp8, Am I right in thinking M34 is also described in the literature as K34? This site already has several names, I think it might be better to stick with K34.**

We saw both acronyms in the literature (Fisher et al., 2009 used M34, Mahli et al 2009 used K34). We though M34 would be easier to differentiate in the paper from K83 (Santarem).

**ln 5 pp 11, suggest rephrasing "equal correlation extends ..."**

Yes thanks, modified. (L12 p9)

**Figure 3 pp 11, Why not also have Taylor plots of monthly data?**

Yes, we could have. Combining Fig.3 (hourly data), Fig. 4 and Fig.5 (that mix hourly/daily and monthly) lead to see the data-model comparison at several time scale, besides because the paper is already long we prefer not to add another figure.

**Figure 4 & 5 I don't find the diel cycles per month that useful or easy to read. I suggest focusing in on just a few key ones that demonstrate the model differences. Can uncertainty be added to the observations?**

Thanks, but as discussed, monthly dial cycles allow us to see a limitation in CAN and CAN-RS hydraulic architecture as it simulates a non-observed mid-day depression on GPP. Uncertainties in flux tower data is known to be about 20% (p7l12 "Flux data are noisy, and *Hollinger and Richardson,* [2005] evaluated the relative uncertainty of H, LE and CO₂ fluxes derived from

eddy-covariance measurements to be around 25% for a temperate site"). However, it is somehow complicated to add them on the figure as it depends on the eddy conditions (and are not provided).

**ln 3-4 pp 17, Can you be sure of this statement, how do you know?**

While local meteorological forcing are hourly (or even half-hourly) and created from measures located closed to the tower, CRU-NCEP data are 6-hourly at a resolution of 0.5°. Therefore, we do expect differences between the two forcing files, that lead to differences in model's outputs.

**Figure 10, Bias seems worst in CAN-RS**

Both models simulate a quite homogeneous pattern of LE over the Amazon (3.2mm over the Amazon), we double checked the code.

**Figure 14, Why is TRUNK not shown?**

The point of this figure (now in SI) is to focus on the soil water uptake influence on the GPP-soil water content relationship. To assess the difference we compared CAN to CAN-RS.

**ln 18-20 pp 19, How are kcmaint and CUE related, from the description it seems CUE = 1 – kcmaint.**

kcmaint indeed plays a large role on the CUE via it's control on Ra,m, but it is not a direct relationship (such as CUE=1-kcmaint). As written in the manuscript: L11p16 "$R_{A,m}$ is calculated for each living compartment as a function of temperature, biomass, prescribed carbon/nitrogen ratio and $k_{cmaint}$, the fraction of allocatable photosynthates consumed for maintenance and growth respiration (which is a tunable parameter, see Table S2)".

**ln 20 pp 22, TRUNK doesn't model diameter size distribution**

Yes, while this part was rewritten, we make sure to be clear in the manuscript. "The simple description of forest demography in CAN does not improve stock and fluxes representation compared to the TRUNK, but it allows to model stand variables (diameter, height distributions) that are just ignored in TRUNK" L16 p18.

Fisher, R. A., C. D. Koven, W. R. L. Anderegg, B. O. Christoffersen, M. C. Dietze, C. E. Farrior, J. A. Holm, G. C. Hurtt, R. G. Knox, P. J. Lawrence, J. W. Lichstein, M. Longo, A. M. Matheny, D. Medvigy, H. C. Muller-Landau, T. L. Powell, S. P. Serbin, H. Sato, J. K. Shuman, B. Smith, A. T. Trugman, T. Viskari, H. Verbeeck, E. Weng, C. Xu, X. Xu, T. Zhang, and P. R. Moorcroft. 2018. Vegetation demographics in Earth System Models: A review of progress and priorities. Global Change Biology 24:35–54.

Goll, D. S., N. Vuichard, F. Maignan, A. Jornet-Puig, J. Sardans, A. Violette, S. Peng, Y. Sun, M. Kva- kic, M. Guimberteau, B. Guenet, S. Zaehle, J. Penuelas, I. Janssens, and P. Ciais. 2017. A representation of the phosphorus cycle for ORCHIDEE (revisionÂaˇ4520). Geosci. Model Dev. 10:3745–3770.

Marthews, T. R., C. A. Quesada, D. R. Gal- braith, Y. Malhi, C. E. Mullins, M. G. Hodnett, and I. Dharssi. 2014. High-resolution hydraulic parameter maps for surface soils in tropical South America. Geosci. Model Dev. 7:711–723. Naudts, K., J. Ryder, M. J. McGrath, J. Otto, Y. Chen, A. Valade, V. Bellasen, G. Berhongaray, G. Bönisch, M. Campioli, J. Ghattas, T. De Groote, V. Haverd, J. Kattge, N. MacBean, F. Maignan, P. Merilä, J. Penuelas, P. Peylin, B. Pinty, H. Pretzsch, E. D. Schulze, D. Solyga, N. Vuichard, Y. Yan, and S. Luyssaert. 2015. A vertically discretised canopy description for ORCHIDEE (SVN r2290) and the modifications to the energy, water and carbon fluxes. Geoscientific Model Development 8:2035–2065.

---

## Author Comment (AC2) · 13 Mar 2019

**Effect of tree demography and flexible root water uptake for modeling the carbon and water cycles of Amazonia**

Emilie Joetzjer[1,2], Fabienne Maignan[1], Jérôme Chave[2], Daniel Goll[1], Ben Poulter[3], Jonathan Barichivich[1,4], Isabelle Maréchaux[2,5], Sebastiaan Luyssaert[1,6], Matthieu Guimberteau[1,7], Kim Naudts[1,8], Damien Bonal[9], Philippe Ciais[1]

[1]Laboratoire des Sciences du Climat et de l'Environnement, LSCE-IPSL (CEA-CNRS-UVSQ), 91190 Gif-sur-Yvette, France
[2]Laboratoire Evolution et Diversité Biologique, UMR 5174, Université Paul Sabatier, CNRS, IRD, 31400 Toulouse, France
[3] NASA Goddard Space Flight Center, Biospheric Sciences Laboratory, Greenbelt, MD, USA
[4] Instituto de Conservación, Biodiversidad y Territorio, Universidad Austral de Chile, Valdivia, Chile, and Center for Climate and Resilience Research, Santiago, Chile
[5] AMAP, Université de Montpellier, IRD, CIRAD, CNRS, INRA, 34000 Montpellier, France
[6] Vrije Universiteit Amsterdam, Faculty of Science, 1081 HV, The Netherlands.
[7]UMR 7619 METIS, Sorbonne Universités, UPMC, CNRS, EPHE, 4 place Jussieu, 75005 Paris, France
[8] Max Planck Institute for Meteorology, Bundesstraβe. 53, 20146 Hamburg, Germany
[9] Université de Lorraine, AgroParisTech, INRA, UMR Silva, 54000 Nancy, France

*Correspondence to*: Emilie.joetzjer@lsce.ipsl.fr

**Abstract.**

Global Dynamic global vegetation models (DGVMs) part of the land surface schemes of Earth System Models do not represent forest structure and demography. Further, those models usually treat water uptake by plants from soil bucket schemes or multiple layers soil models, with non-physical water stress functions. This situation is a source of uncertainty for modelling the current state and future dynamics of Amazonian forest. We included recruitment processes and added a physically based model of soil water limitation on root water uptake for tropical evergreen forests in ORCHIDEE-CAN, a DGVM with demography equations previously established for temperate and boreal forests. The model was re-calibrated against tropical forest inventory measurements in Guyana and Brazil, including biomass-age relationships, diameter and height distributions. We compared the output of two model versions, one with forest demography only (CAN) and the other with demography and hydraulic transport of water from soil pores to roots coupled to xylem conduction by stems (CAN-RS). The results of the CAN and CAN-RS versions are compared with those of the standard TRUNK version of the DGVM used for global applications, which follows a big-leaf approximation, does not resolve forest demography and has a constant mortality applied to a well-mixed biomass pool. Site-level observations of turbulent energy and $CO_2$ fluxes at flux tower locations are used to evaluate the three versions of the model, as well as measurements of carbon stocks and stand density at inventory plots. Gridded observation-based models output of photosynthesis (GPP) and evapotranspiration (LE) are used across the Amazon basin. CAN and CAN-RS can reproduce observed forest structure variables, including tree density and height / diameters distribution, which were not modeled by TRUNK. In CAN-RS, water uptake from tree roots sustained from the deepest soil layers during the dry season, significantly improved the modeling of seasonal photosynthesis and evapotranspiration variations compared to CAN, especially over the Guyana and Brazilian shields with prevailing clay soils. The performances of CAN-RS are shown to be equivalent to those of TRUNK for carbon and water fluxes and carbon stocks across Amazonia, despite new processes being added. This result indicates that forest demography and mechanistic root water uptake did not degrade the DGVM performance while offering a more realistic simulation of key processes for the Amazon forest.

**1 Introduction**

Despite the importance of the Amazonian rainforests for global climate [*Eltahir and Bras*, 1994; *Werth and Avissar*, 2002] large uncertainties impede the production of robust future projections of changes in net carbon uptake over Amazonia
5   [*Poulter et al.*, 2010; *Arora et al.*, 2013; *Jones et al.*, 2013]. An analysis of variance on simulation outputs from 12 Earth System models (ESM) showed that uncertainties in projections of terrestrial carbon uptake are primarily driven by model structure [*Lovenduski and Bonan*, 2017]. These uncertainties arise from both the atmospheric [*Ahlström et al.*, 2012] and the land surface components [*Booth et al.*, 2012; *Sitch et al.*, 2015]. In land models (dynamic global vegetation models, or DGVMs) large sources of uncertainty include the vegetation response to droughts [*Restrepo-Coupe et al.*, 2016], and tree
10   demographic processes [*Fisher et al.*, 2010; *Rödig et al.*, 2018]. Most DGVMs simulate the effect of water shortage on plant functioning by lowering leaf gas exchange rates using a multiplicative water stress factor that depends on soil moisture [*Christoffersen et al.*, 2014] and by including atmospheric water stress from increased vapour pressure deficit in their parameterization of stomatal conductance. With this simplification, models typically fail to capture tropical carbon and water flux seasonality [*Poulter et al.*, 2009; *Restrepo-Coupe et al.*, 2016], and vegetation response to drought [*Powell et al.*, 2013;
15   *Joetzjer et al.*, 2014]. A few global DGVMs have recently adopted a more explicit representation of the soil-plant-atmosphere water column [*Bonan et al.*, 2014; *Christoffersen et al.*, 2014; *Xu et al.*, 2016], but much research is still needed to fully model these processes.

In most DGVMs, water availability in the root zone is quantified using the root biomass-weighted or root profile-weighted
20   sum of soil layer moisture. Yet, this model structure overlooks the observation that soil-to-root water flow depends on soil and root hydraulic properties, which vary in time and space [*Sperry et al.*, 2002]. A prevailing assumption is that the upper soil layers, with higher root biomass, contribute more to soil water uptake. This however overlooks the fact that tree water potentials preferentially equilibrate with the wettest part of the soil [*Schmidhalter*, 1997], a process controlled not only by the density of roots but also by the soil-to-root resistance. In turn, the soil-to-root resistance is non-linearly related to soil
25   water content [*Gardner*, 1960]. Overall, this approach leads to an overestimation of the water stress experienced by trees. We evaluated the effect of representing layer-to-layer heterogeneity of soil water availability for trees and test if this process improves DGVMs carbon and water fluxes representation. Besides, first-generation DGVMs that are based on a spatially aggregated hypothesis, also called "big-leaf" models, are progressively being superseded by second generation DGVMs (2gDGVM). This new generation of models is partly inspired by individual plant-based and forest stand models (e.g., [*Fyllas*
30   *et al.*, 2014; *Fischer et al.*, 2016; *Maréchaux and Chave*, 2017]), and they explicitly represent forest dynamics via tree demography (cohort-based) and vertical competition for light. There are several approaches to represent tree demography in models. In some models forest structure and tree demography emerge from a mechanistic representation of competition and recruitment schemes with different species or groups of species having different functional traits. This method has given insights on tropical forests dynamics and resilience [*Zhang et al.*, 2015; *Levine et al.*, 2016; *Xu et al.*, 2016; *Fisher et al.*,
35   2018]. However they come with complex parameterizations, high computational cost and are not practical to run at global scale. In this study, we propose to evaluate and test the benefit of an intermediate complexity second generation DGVM that represents demography and forest structure by downscaling NPP into several size classes and simulating mortality from tree density exceeding a threshold by killing preferentially a fraction of the smaller size classes, based on self-thinning principles. Self-thinning is here parameterized but has been observed as an emerging property in tropical stands [*Pillet et al.*, 2017].
40
This study explores the importance of tree demographic and below-ground hydraulic processes on the Amazonian carbon

and water cycles, by testing the performances of three versions of the same model: the ORCHIDEE DGVM (Organizing Carbon and Hydrology in Dynamic Ecosystems). As a reference version, we used the TRUNK version updated for the CMIP6 exercise (Peylin et al., *in prep*) and widely used in global carbon cycle studies. TRUNK is based on Krinner et al. (2005) uses a "big-leaf" approximation, and simulates biomass dynamics from the allocation of NPP to leaves, wood and roots with a constant mortality. A single well-mixed pool thus describes woody biomass in the TRUNK. Water uptake by roots in TRUNK is calculated by weighting a static root profile discretized upon 11 soil layers by soil moisture in each layer. CAN has the same photosynthesis model than TRUNK but includes a simplified forest demography model with 20 diameter classes upon which stand level GPP is distributed unevenly to favour high diameters, and mortality being the result of light competition (self-thinning). CAN-RS is equal to CAN but has a new root water uptake model described in this study.

We evaluated (1) the effect of representing layer-to-layer heterogeneity of soil water availability for trees and test if this process improves DGVMs carbon and water fluxes representation by comparing CAN-RS and CAN (2) the effect of representing demography and forest structure by distributing NPP into several classes of diameter, including self-thinning mortality and recruitment by comparing CAN and TRUNK and (3) the combined effect of including layer-to-layer heterogeneity of soil water and forest demography by comparing CAN-RS with TRUNK. By comparing simulations by these three versions over Amazonia against forest inventory data for biomass and stand characteristics, local site level flux tower measurements of carbon and latent heat exchange, and regional water and carbon fluxes and stocks observation-based models, 
[revised manuscript text omitted]
 [*Mualem*, 1976; *van Genuchten*, 1980]. For example, they simulate a non-observed midday depression (Fig. 5), indicating that the above-ground hydraulic path needs to be revisited. It is well known that changes in the spatial resolution of the soil input data by aggregating small-scale information causes serious problems in models [*Van Looy et al.*, 2017], as well as the use of coarse soil texture classes [*Kishné et al.*, 2017]. Thus, along with improving model representation of the hydraulic gradient from the soil to the plant in DGVMs [this study, *Sperry et al.*,
20    2002; *Fisher et al.*, 2006], it is important to improve the parameterization of the physical soil environment [*Marthews et al.*, 2014].

**4.2 Modeling forest structure and demography**

25    As with most DGVMs [*Castanho et al.*, 2015; *Johnson et al.*, 2016], TRUNK, CAN and CAN-RS fail to capture the SW-NE gradient of AGB (and BA) across Amazonia (Fig. 12) — they simulate a quasi-constant AGB across the basin (Fig. S5). This AGB gradient could be caused either by productivity or tree mortality. Spatial variation in wood productivity can be linked to spatial variability in soil properties [*Quesada et al.*, 2012], like soil fertility [*ter Steege et al.*, 2006 ; *Malhi et al.*, 2004, Turner et al., 2018] and soil hydraulic parameters. Therefore, the incorporation of detailed soil hydraulic parameters
30    maps (e.g., [*Marthews et al.*, 2014]) and inclusion of nutrient cycles into ORCHIDEE would yield advances. Besides, and because of the negative relationship between soil fertility and wood density [*Baker et al.*, 2004; *ter Steege et al.*, 2006; *Patiño et al.*, 2009], and wood density and tree mortality [*King et al.*, 2006]; wood density should vary across the basin (rather than using a single parameter for the evergreen tropical forest PFT); this requires the use of wood density maps [*ter Steege et al.*, 2006]. Finally, mortality processes need to be linked to edaphic properties.
35

   Besides variation in productivity, it has been shown that variation in tree mortality is a key driver of AGB across Amazonia [*Johnson et al.*, 2016 and references within]. TRUNK, CAN and CAN-RS show a quasi-linear positive relationship between NPP and AGB, despite their differences in forest representation and tree mortality scheme over Amazonia (Fig. 13a) and when focusing on different regions of the basin (Figs. S8 and S9). For high NPP, CAN-RS and CAN showed a saturation of
40    GPP (not shown) and AGB around 250 tC ha$^{-1}$ (Fig. 13a) due to a breakpoint in precipitation at 2000 mm yr$^{-1}$ (Fig. 13b), this phenomenon was also identified by Alstrom et al., 2016. CAN-RS has similar performance than the big-leaf / single biomass pool TRUNK version to represent forest fluxes, stocks and dynamics (Figs. S1,S3,S5,12,13), despite its higher complexity and more realistic description of soil water uptake and demography. This is a positive result since a model like TRUNK is easier to adjust and intrinsically more stable, but less informative. For instance, adjusting the constant mortality parameter in

TRUNK allows to match the observed biomass well, but lacks a mechanistic basis. CAN-RS offers the advantage to be assessable against other key measurable forest stand variables than total biomass, such as height and diameter classes for different forest ages (e.g. *Joetzjer et al.*, [2017]). Nevertheless, since demography parameters in CAN-RS are set constant for a single PFT describing all evergreen tropical forests, spatial variability of growth rates and mortality across the Amazon remains rather uniform compared to observations. Additional processes such as climate driven mortality and nutrient (phosphorus) limitation on growth leading to the prevalence of species with different functional traits across the Amazon would need to be included in the future development of this model.

[Figure]

**Figure 13. (a) scatter plots of mean AGB from 1981 to 2016 plotted against mean annual NPP and (b) annual precipitation averaged over the same time period for Amazonia.**

**4.3. Concluding remarks**

The simple description of forest demography in CAN does not improve stock and fluxes representation compared to the TRUNK, but it allows to model stand variables (diameter, height distributions) that are just ignored in TRUNK. We do not expect an automatic improvement of the performance of CAN-RS against TRUNK and we view as positive result that the more realistic CAN version has in fact similar performance than the easy-to-tune TRUNK version, while allowing to evaluate the CAN output against a new set of measurable variables that are known to be important to understand forest carbon dynamics. This approach is similar to C. Prentice (2015)'s remark: *"Although it seems reasonable to expect that a model including a larger subset of processes that are known to be important should be more realistic than a simpler model, increases in reliability and robustness are by no means automatic"*, which justifies the inclusion of known important processes and calls for observations to evaluate those new processes. Concerning the root water uptake scheme, CAN-RS and TRUNK show similar performances. However, TRUNK uses a simplistic formulation of soil water stress on stomatal conductance through Vcmax (see section D in SI), while in CAN water supply is calculated via a more realistic hydraulic architecture inspired from Hickler et al. (2006). Having a mechanistic representation of soil water stress is crucial for DGVMs to understand the vegetation response to drought. Our point here is to show that in normal conditions, CAN-RS performs as well as the TRUNK, but because of its mechanistic approach CAN-RS has more potential, for instance to simulate hydraulic failure (and possibility mortality occurring from it). Comparing CAN to CAN-RS, showed that trees preferentially use water in the deepest soil layer during the dry season which led to improve LE and GPP seasonality for the soil types with a constraining soil water retention curve. Thus, because of a more realistic representation of forest structure and root water uptake than TRUNK, CAN-RS should be used in future studies.

**Aknowledgments**

Data acquisition in French Guiana was supported by an "investissement d'avenir" grant from the Agence Nationale de la Recherche (CEBA, ref ANR-10-LABX-25-01). J.B. acknowledges support from (CR)[2] Chile (CONICYT/FONDAP/15110009). Matthieu Guimberteau, D. Goll and P. Ciais are funded by the European Research Council Synergy grant ERC-2013-SyG-610028 IMBALANCE-P. We also acknowledge the European Union Climate KIC grant FOREST Specific Grant Agreement EIT/CLIMATE KIC/SGA2016/1CNES (TOSCA program) for funding.

**Code availability**

The code of ORCHIDEE-CAN r2290 (Naudts et al., 2015) can be accessed from http://dx.doi.org/10.14768/06337394-73A9-407C-9997-0E380DAC5595

---

## Author Comment (AC3) · 13 Mar 2019

**Effect of  tree demography and flexible root water uptake for modelling the carbon and water cycles of Amazonia**

Emilie Joetzjer[1,2], Fabienne Maignan[1], Jérôme Chave[2], Daniel Goll[1], Ben Poulter[3], Jonathan Barichivich[1,4], Isabelle Maréchaux[2,5], Sebastiaan Luyssaert[1,6], Matthieu Guimberteau[1,7], Kim Naudts[1,8], Damien Bonal[9], Philippe Ciais[1]

[1]Laboratoire des Sciences du Climat et de l'Environnement, LSCE-IPSL (CEA-CNRS-UVSQ), 91190 Gif-sur-Yvette, France

[2]Laboratoire Evolution et Diversité Biologique, UMR 5174, Université Paul Sabatier, CNRS, IRD, 31400 Toulouse, France

[3]NASA Goddard Space Flight Center, Biospheric Sciences Laboratory, Greenbelt, MD, USA

[4] Instituto de Conservación, Biodiversidad y Territorio, Universidad Austral de Chile, Valdivia, Chile, and Center for Climate and Resilience Research, Santiago, Chile

[5] AMAP, Université de Montpellier, IRD, CIRAD, CNRS, INRA, 34000 Montpellier, France

[6] Vrije Universiteit Amsterdam, Faculty of Science, 1081 HV, The Netherlands.

[7]UMR 7619 METIS, Sorbonne Universités, UPMC, CNRS, EPHE, 4 place Jussieu, 75005 Paris, France

[8] Max Planck Institute for Meteorology, Bundesstraβe. 53, 20146 Hamburg, Germany

[9] Université de Lorraine, AgroParisTech, INRA, UMR Silva, 54000 Nancy, France

*Correspondence to*: Emilie.joetzjer@lsce.ipsl.fr

**Abstract.** ~~Amazonian forest plays a crucial role in regulating the carbon and water cycles in the global climate system. However, the representation of biogeochemical fluxes and forest structure in dynamic global vegetation models (DGVMs) remains challenging. This situation has considerable implications for modelling the state and dynamics of Amazonian forest. To address these limitations, we present an adaptation of the ORCHIDEE CAN DGVM, a second-generation DGVM that explicitly models tree demography and canopy structure with an allometry-based carbon allocation scheme and accounts for hydraulic architecture in the soil-stem-leaf continuum. We use two versions of this DGVM: the first one (CAN) includes a new parameterization for Amazonian forest; the second one (CAN-RS) additionally includes a mechanistic root water uptake module, which models the hydraulic resistance of the water transfer from soil pores to roots. We compared the results with the simulation output of the "big leaf" standard version of the ORCHIDEE DGVM (TRUNK) and with observations of turbulent energy and CO₂ fluxes at flux tower locations, of carbon stocks and stand density at inventory plots and observation-based models of photosynthesis (GPP) and evapotranspiration (LE) across the Amazon basin. CAN-RS reproduced observed carbon and water fluxes and carbon stocks as well as TRUNK across Amazonia, both at local and at regional scales. In CAN-RS, water uptake by tree roots in the deepest soil layers during the dry season significantly improved the modelling of GPP and LE seasonal cycles, especially over the Guianan and Brazilian Shields. While CAN-RS correctly captures forest structure, CAN-RS does not outperform TRUNK to reproduced observed carbon and water fluxes and carbon stocks across Amazonia, both at local and at regional scales. These results imply that explicit coupling of the water and carbon cycles improves the representation of biogeochemical cycles in Amazonia and their spatial variability. Representing the variation in the ecological functioning of Amazonia should be the next step to improve the performance and predictive ability of new generation DGVMs.~~

Global Dynamic global vegetation models (DGVMs) part of the land surface schemes of Earth System Models do not represent forest structure and demography. Further, those models usually treat water uptake by plants from soil bucket schemes or multiple layers soil models, with non-physical water stress functions. This situation is a source of uncertainty for modelling the current state and future dynamics of Amazonian forest. We included recruitment processes and added a physically based model of soil water limitation on root water uptake for tropical evergreen forests in ORCHIDEE-CAN, a DGVM with demography equations previously established for temperate and boreal forests. The model was re-calibrated against tropical forest inventory measurements in Guyana and Brazil, including biomass-age relationships, diameter and height distributions. We compared the output of two model versions, one with forest demography only (CAN) and the other with demography and hydraulic transport of water from soil pores to roots coupled to xylem conduction by stems (CAN-RS). The results of the CAN and CAN-RS versions are compared with those of the standard TRUNK version of the DGVM used for global applications, which follows a big-leaf approximation, does not resolve forest demography and has a constant mortality applied to a well-mixed biomass pool. Site-level observations of turbulent energy and $CO_2$ fluxes at flux tower locations are used to evaluate the three versions of the model, as well as measurements of carbon stocks and stand density at inventory plots. Gridded observation-based models output of photosynthesis (GPP) and evapotranspiration (LE) are used across the Amazon basin. CAN and CAN-RS can reproduce observed forest structure variables, including tree density and height / diameters distribution, which were not modeled by TRUNK. In CAN-RS, water uptake from tree roots sustained from the deepest soil layers during the dry season, significantly improved the modeling of seasonal photosynthesis and evapotranspiration variations compared to CAN, especially over the Guyana and Brazilian shields with prevailing clay soils. The performances of CAN-RS are shown to be equivalent to those of TRUNK for carbon and water fluxes and carbon stocks across Amazonia, despite new processes being added. This result indicates that forest demography and mechanistic root water uptake did not degrade the DGVM performance while offering a more realistic simulation of key processes for the Amazon forest.

**1 Introduction**

Amazonian rainforests store approximately half of the world's tropical forest carbon stock [*Baccini et al.*, 2012] and play a crucial role in global water, energy and carbon cycling [*Eltahir and Bras*, 1994; *Werth and Avissar*, 2002]. The resilience and resistance of these forests to climate change is of great concern, especially since a significant portion of Amazonia will likely experience longer and drier dry seasons by the end of the 21st century [*Joetzjer et al.*, 2013; *Boisier et al.*, 2015]. Future changes in the rate of carbon sequestered by Amazonia could potentially lead the global climate system to a critical tipping point [*Lenton et al.*, 2008; *Nobre and Borma*, 2009; *Ahlström et al.*, 2017], and trigger positive carbon cycle climate feedbacks from forest dieback. Yet, large uncertainties impede the production of robust future projections of changes in net carbon uptake over Amazonia [*Poulter et al.*, 2010; *Arora et al.*, 2013; *Jones et al.*, 2013] – current model projections range from no change, or an increase in tree biomass production [*Rammig et al.*, 2010; *Cox et al.*, 2013; *Huntingford et al.*, 2013], to large-scale Amazonian dieback [*Cox et al.*, 2004; *Good et al.*, 2011].

Despite the importance of the Amazonian rainforests for global climate [*Eltahir and Bras*, 1994; *Werth and Avissar*, 2002] large uncertainties impede the production of robust future projections of changes in net carbon uptake over Amazonia [*Poulter et al.*, 2010; *Arora et al.*, 2013; *Jones et al.*, 2013]. An analysis of variance on simulation outputs from 12 Earth System models (ESM) showed that uncertainties in projections of terrestrial carbon uptake are primarily driven by model structure [*Lovenduski and Bonan*, 2017]. These uncertainties arise from both the atmospheric [*Ahlström et al.*, 2012] and the land surface components [*Booth et al.*, 2012; *Sitch et al.*, 2015]. In land models (dynamic global vegetation models, or DGVMs) large sources of uncertainty include the vegetation response to droughts [*Restrepo-Coupe et al.*, 2016], and tree

demographic processes [*Fisher et al.*, 2010; *Rödig et al.*, 2018]. Most DGVMs simulate the effect of water shortage on plant functioning by lowering leaf gas exchange rates using a multiplicative water stress factor that depends on soil moisture [*Christoffersen et al.*, 2014] and by including atmospheric water stress from increased vapour pressure deficit in their parameterization of stomatal conductance. With this simplification, models typically fail to capture tropical carbon and water

5    flux seasonality [*Poulter et al.*, 2009; *Restrepo-Coupe et al.*, 2016], and vegetation response to drought [*Powell et al.*, 2013; *Joetzjer et al.*, 2014]. A few global DGVMs have recently adopted a more explicit representation of the soil-plant-atmosphere water column [*Bonan et al.*, 2014; *Christoffersen et al.*, 2014; *Xu et al.*, 2016], but much research is still needed to fully model these processes.

10    In most DGVMs, water availability in the root zone is quantified using the root biomass-weighted or root profile-weighted sum of soil layer moisture. Yet, this model structure overlooks the observation that soil-to-root water flow depends on soil and root hydraulic properties, which vary in time and space [*Sperry et al.*, 2002]. A prevailing assumption is that the upper soil layers, with higher root biomass, contribute more to soil water uptake. This however overlooks the fact that tree water potentials preferentially equilibrate with the wettest part of the soil [*Schmidhalter*, 1997], a process controlled not only by

15    the density of roots  but also by the soil-to-root resistance. In turn, the soil-to-root resistance is non-linearly related to soil water content [*Gardner*, 1960]. Overall, this approach leads to an overestimation of the water stress experienced by trees. We evaluated the effect of representing layer-to-layer heterogeneity of soil water availability for trees and test if this process improves DGVMs carbon and water fluxes representation. Besides, first-generation "" DGVMs that are based on a spatially aggregated hypothesis, also called "big-leaf" models, are progressively being superseded by second

20    generation DGVMs (2gDGVM). This new generation of models is partly inspired by individual plant-based and forest stand models (e.g., [*Fyllas et al.*, 2014; *Fischer et al.*, 2016; *Maréchaux and Chave*, 2017]), and they explicitly represent forest dynamics via tree demography (cohort-based) and vertical competition for light. There are several approaches to represent tree demography in models. In some models forest structure and tree demography emerge from a mechanistic representation of competition and recruitment schemes with different species or groups of species having different functional traits. This

25    method has given insights on tropical forests dynamics and resilience [*Zhang et al.*, 2015; *Levine et al.*, 2016; *Xu et al.*, 2016; *Fisher et al.*, 2018]. However they come with complex parameterizations, high computational cost and are not practical to run at global scale. In this study, we propose to evaluate and test the benefit of an intermediate complexity second generation DGVM that represents demography and forest structure by downscaling NPP into several size classes and simulating mortality from tree density exceeding a threshold by killing preferentially a fraction of the smaller size classes,

30    based on self-thinning principles. Self-thinning is here parameterized but has been observed as an emerging property in tropical stands [*Pillet et al.*, 2017].

This study explores the  importance of tree demographic  and below-ground hydraulic processes on the Amazonian carbon and water cycles, by testing the performances of three versions of the same

35    model: the ORCHIDEE DGVM (Organizing Carbon and Hydrology in Dynamic Ecosystems). As a reference version, we used the TRUNK version updated for the CMIP6 exercise (Peylin et al., *in prep*) and widely used in global carbon cycle studies. TRUNK is based on Krinner et al. (2005) uses a "big-leaf" approximation, and simulates biomass dynamics from the allocation of NPP to leaves, wood and roots with a constant mortality. A single well-mixed pool thus describes woody biomass in the TRUNK. Water uptake by roots in TRUNK is calculated by weighting a static root profile discretized upon 11

40    soil layers by soil moisture in each layer. CAN has the same photosynthesis model than TRUNK but includes a simplified forest demography model with 20 diameter classes upon which stand level GPP is distributed unevenly to favour high diameters, and mortality being the result of light competition (self-thinning). CAN-RS is equal to CAN but has a new root water uptake model described in this study.

We evaluated (1) the effect of representing layer-to-layer heterogeneity of soil water availability for trees and test if this process improves DGVMs carbon and water fluxes representation by comparing CAN-RS and CAN (2) the effect of representing demography and forest structure by distributing NPP into several classes of diameter, including self-thinning mortality and recruitment by comparing CAN and TRUNK and (3) the combined effect of including layer-to-layer heterogeneity of soil water and forest demography by comparing CAN-RS with TRUNK. By comparing simulations by these three versions over Amazonia against forest inventory data for biomass and stand characteristics, local site level flux tower measurements of carbon and latent heat exchange, and regional water and carbon fluxes and stocks observation-based models, our results shed light on critical processes whose explicit representation would help to improve the performance of 2gDGVMs in general, and CAN in particular, and enhance their predictive ability on the fate of the largest tropical forest on Earth.

**2. Methods**

**2.1 Model description and experimental design**

**2.1.1 General model description**

[revised manuscript text omitted]
 [*Mualem*, 1976; *van Genuchten*, 1980]. For example, they simulate a non-observed midday depression (Fig. 5), indicating that the above-ground hydraulic path needs to be revisited. It is well known that changes in the spatial resolution of the soil input data by aggregating small-scale information causes serious problems in models [*Van Looy et al.*, 2017], as well as the use of coarse soil texture classes [*Kishné et al.*, 2017]. Thus, along with improving model representation of the hydraulic gradient from the soil to the plant in DGVMs [this study, *Sperry et al.*, 2002; *Fisher et al.*, 2006], it is important to improve the parameterization of the physical soil environment [*Marthews et al.*, 2014].
* * *
**4.2 Modeling forest structure and demography**

[revised manuscript text omitted]

The simple description of forest demography in CAN does not improve stock and fluxes representation compared to the TRUNK, but it allows to model stand variables (diameter, height distributions) that are just ignored in TRUNK. We do not expect an automatic improvement of the performance of CAN-RS against TRUNK and we view as positive result that the more realistic CAN version has in fact similar performance than the easy-to-tune TRUNK version, while allowing to evaluate the CAN output against a new set of measurable variables that are known to be important to understand forest carbon dynamics. This approach is similar to C. Prentice (2015)'s remark: *"Although it seems reasonable to expect that a model including a larger subset of processes that are known to be important should be more realistic than a simpler model, increases in reliability and robustness are by no means automatic",* which justifies the inclusion of known important processes and calls for observations to evaluate those new processes. Concerning the root water uptake scheme, CAN-RS and TRUNK show similar performances. However, TRUNK uses a simplistic formulation of soil water stress on stomatal conductance through Vcmax (see section D in SI), while in CAN water supply is calculated via a more realistic hydraulic architecture inspired from Hickler et al. (2006). Having a mechanistic representation of soil water stress is crucial for DGVMs to understand the vegetation response to drought. Our point here is to show that in normal conditions, CAN-RS performs as well as the TRUNK, but because of its mechanistic approach CAN-RS has more potential, for instance to simulate hydraulic failure (and possibility mortality occurring from it). Comparing CAN to CAN-RS, showed that trees preferentially use water in the deepest soil layer during the dry season which led to improve LE and GPP seasonality for the soil types with a constraining soil water retention curve. Thus, because of a more realistic representation of forest structure and root water uptake than TRUNK, CAN-RS should be used in future studies.

**Aknowledgments**

Data acquisition in French Guiana was supported by an "investissement d'avenir" grant from the Agence Nationale de la Recherche (CEBA, ref ANR-10-LABX-25-01). J.B. acknowledges support from (CR)$^2$ Chile (CONICYT/FONDAP/15110009). Matthieu Guimberteau, D. Goll and P. Ciais are funded by the European Research Council Synergy grant ERC-2013-SyG-610028 IMBALANCE-P. We also acknowledge the European Union Climate KIC grant FOREST Specific Grant Agreement EIT/CLIMATE KIC/SGA2016/1CNES (TOSCA program) for funding.

**Code availability**

5    The code of ORCHIDEE-CAN r2290 (Naudts et al., 2015) can be accessed from http://dx.doi.org/10.14768/06337394-73A9-407C-9997-0E380DAC5595